# Rethinking Spectral Graph Neural Networks with Spatially Adaptive Filtering

## Abstract

Whilst spectral Graph Neural Networks (GNNs) are theoretically well-founded in the spectral domain, their practical reliance on polynomial approximation implies a profound linkage to the spatial domain. As previous studies rarely examine spectral GNNs from the spatial perspective, their spatial-domain interpretability remains elusive, *e.g.*, what information is essentially encoded by spectral GNNs in the spatial domain? In this paper, to answer this question, we establish a theoretical connection between spectral filtering and spatial aggregation, unveiling an intrinsic interaction that spectral filtering implicitly leads the original graph to an adapted new graph, explicitly computed for spatial aggregation. Both theoretical and empirical investigations reveal that the adapted new graph not only exhibits non-locality but also accommodates signed edge weights to reflect label consistency between nodes. These findings thus highlight the interpretable role of spectral GNNs in the spatial domain and inspire us to rethink graph spectral filters beyond the fixed-order polynomials, which neglect global information. Built upon the theoretical findings, we revisit the state-of-the-art spectral GNNs and propose a novel Spatially Adaptive Filtering (SAF) framework, which leverages the adapted new graph by spectral filtering for an auxiliary non-local aggregation. Notably, our proposed SAF comprehensively models both node similarity and dissimilarity from a global perspective, therefore alleviating persistent deficiencies of GNNs related to long-range dependencies and graph heterophily. Extensive experiments over 13 node classification benchmarks demonstrate the superiority of our proposed framework to the state-of-the-art models.

## 1 Introduction

Graph Neural Networks (GNNs) have shown remarkable abilities to uncover the intricate dependencies within graph-structured data, and achieved tremendous success in graph machine learning (Wu et al., 2021; Chami et al., 2022; Gao et al., 2023). Spectral GNNs are a class of GNNs rooted in spectral graph theory (Chung, 1996; Shuman et al., 2013), implementing graph convolutions via spectral filters (Defferrard et al., 2016; Kipf & Welling, 2017). Whilst various spectral filtering strategies have been proposed for spectral GNNs, their practical implementations always involve approximating graph filters with fixed-order polynomials for computational efficiency (Wang & Zhang, 2022; He et al., 2022). This truncated approach essentially relies on the direct extraction of spatial features from the local regions of nodes. As such, the spatial domain of a graph, albeit loosely connected to spectral GNNs in theory, still plays a crucial role in effectively learning node representations.

However, there is a notable lack of research examining spectral GNNs from the spatial perspective. Though recent studies analyze both spectral and spatial GNNs to elucidate their similarities in model formulations (Balcilar et al., 2020; Chen et al., 2021; Guo & Wei, 2023b), outcomes (Ma et al., 2021; Zhu et al., 2021), and expressiveness (Chen et al., 2020; Balcilar et al., 2021; Wang & Zhang, 2022; Sun et al., 2023), they ignore exploring the interpretability that could arise mutually from the other domain. Specifically, whilst most spectral GNNs have well explained their learned graph filters in the spectral domain (He et al., 2021; Wang & Zhang, 2022; He et al., 2022; Guo et al., 2023), understandings from the spatial viewpoint are merely limited to fusing multi-scale graph information (Liao et al., 2019); this unfortunately lacks a deeper level of interpretability in the vertex domain. Therefore, a natural question arises: *what information is essentially encoded by spectral GNNs in the spatial domain?*

In this work, we attempt to answer this question by exploring the connection between spectral filtering and spatial aggregation. The former is the key component in spectral GNNs, while the latter is closely associated with spatial GNNs utilizing recursive neighborhood aggregation. In existing GNN frameworks, these two approaches rarely interact each other at the risk of domain information trade-offs due to uncertainty principles (Heisenberg, 1927; Folland & Sitaram, 1997; Agaskar & Lu, 2013). Recognizing the spatial significance in spectral filtering, He et al. (2021) have recently considered non-negative constraints as part of a generalized graph optimization problem. Notably, however, spatial aggregation meanwhile resembles the optimizing trajectory of the same optimization problem through iterative steps, which may be easily overlooked. Inspired by such observation, we examine, for the first time, the theoretical interaction between spectral filtering and spatial aggregation. This exploration has led us to uncover an intriguing theoretical interplay, *i.e.*, spectral filtering implicitly modifies the original graph, transforming it into a new one that explicitly functions as a computation graph for spatial aggregation. Delving deeper, we discover that the adapted new graph enjoys some desirable properties, enabling a direct link among nodes that originally require multiple hops to do so, thereby exhibiting nice non-locality. Moreover, we find that the new graph edges allow signed weights, which turns out capable of distinguishing between label agreement and disagreement of the connected nodes.

Overall, these findings underscore the interpretable role and significance of spectral GNNs in the spatial domain, inspiring us to rethink graph spectral filters beyond the fixed-order polynomials, which confine models' receptive fields and overlook global information. Concretely, we propose a novel Spatially Adaptive Filtering (SAF) framework, for fully exploring spectral GNNs in the spatial domain. SAF leverages the adapted new graph by spectral filtering for auxiliary spatial aggregation and allows individual nodes to flexibly balance between spectral and spatial features. By performing non-local aggregation with signed edge weights, our SAF adeptly overcomes the limitations of truncated polynomials, enabling the model to capture both node similarity and dissimilarity at a global scale. As a benefit, it can mitigate persistent deficiencies of GNNs regarding long-range dependencies and graph heterophily. The contributions are summarized as follows:

- Our investigation into spectral GNNs in the spatial domain reveals that graph spectral filtering fundamentally alters the original graph, imbuing it with non-locality and signed edge weights that discern label consistency among nodes.

- We further propose Spatially Adaptive Filtering (SAF), a paradigm-shifting approach to spectral GNNs that jointly leverages graph learning in both spatial and spectral domains, making it a powerful tool for capturing long-range dependencies and handling graph heterophily.

- We showcase SAF framework on BernNet, a leading spectral GNN model with theoretical support. Extensive experiments over 13 node classification benchmarks exhibit notable improvements of up to 15.37%, and show that our model beats the best-performing spectral GNNs on average.

## 2 RELATED WORKS

**Graph Neural Networks.** GNNs can be broadly divided into spatial and spectral GNNs. Spatial GNNs leverage the spatial connections among nodes to perform message passing, also known as spatial aggregation (Gilmer et al., 2017; Hamilton et al., 2017). For a thorough review of spatial GNNs, we direct readers to the works (Zhou et al., 2020; Wu et al., 2021). Spectral GNNs leverage the graph's spectral domain for convolution or, alternatively, spectral filtering (Defferrard et al., 2016; Kipf & Welling, 2017). Prevailing approaches focus on developing polynomial graph filters, by either learning polynomial coefficients (Levie et al., 2018; Dong et al., 2021; Chien et al., 2021; He et al., 2021; Ju et al., 2022; He et al., 2022; Wang & Zhang, 2022; Guo et al., 2023) or concurrently optimizing the polynomial basis for better real-world adaption (Tao et al., 2023; Guo & Wei, 2023a). While these methods are theoretically grounded in the spectral domain, their practical reliance on polynomial approximation hints at a profound linkage to the spatial domain. However, the spatial-domain interpretation of spectral GNNs is rarely examined. To this end, we delve into in this paper the intrinsic information spectral GNNs convey within the spatial context.

**Unified Viewpoints for GNNs.** Several works have explored the nuances between spatial and spectral GNNs. Early studies by Balcilar et al. (2020) and Chen et al. (2021) examined their similarities in model formulations. Chen et al. (2020) proved their spatial GNN's anti-oversmoothing ability via spectral analysis. Ma et al. (2021) and Zhu et al. (2021) utilized the graph signal denoising problem

to integrate both GNN types. Balcilar et al. (2021) and Wang & Zhang (2022) further explored their expressiveness equivalence. Recently, Sun et al. (2023) have highlighted the feature space constraints of both spatial and spectral GNNs, while Guo & Wei (2023b) attempted to combine them via a residual connection module. Though these studies effectively bridge spectral and spatial GNNs, they remain focused on congruencies. Unlike them, our work uniquely investigates the interpretability of spectral GNNs in the spatial domain, emphasizing the synergy between spectral filtering and spatial aggregation. The empirical success of our proposed method (see Section 6), stemming from this in-depth analysis, further underscores our practical contributions to the literature.

**Long-range Dependencies.** While substantial efforts have been directed towards capturing long-range dependencies in spatial GNNs (Gasteiger et al., 2019; Chen et al., 2020; Eliasof et al., 2021; Pei et al., 2020; Liu et al., 2021; Wu et al., 2022), the exploration of the same challenge in spectral GNNs remains under-studied. To fill this gap, we propose a SAF framework, which emerges as a valuable consequence of analyzing spectral GNNs in the spatial domain, enhancing their long-range dependency capture. Concurrently, Bo et al. (2023) also introduced Specformer to addresses long-range dependencies for spectral GNNs, using a Transformer based set-to-set spectral filter. However, it lacks spatial-domain interpretability and introduce more trainable parameters. In contrast, our approach creates a non-local new graph without learning additional parameters, simultaneously elucidating the interpretive implications of spectral GNNs in the spatial domain.

**Graph Heterophily.** Graph heterophily (Pei et al., 2020; Zhu et al., 2020a), where different labeled nodes connect, challenges GNNs operating under the homophily assumption (McPherson et al., 2001). Although many GNNs have been crafted to manage heterophilic connections (see further remarks in Appendix C.2), our proposed SAF stands out in addressing graph heterophily. Specifically, SAF innovatively conducts an auxiliary non-local aggregation using signed edge weights, emphasizing both intra-class similarity and inter-class difference on a global scale. One should note that a recent work (Li et al., 2022) bear some resemblance to ours, introducing GloGNN and GloGNN++ to capture global homophily beyond immediate neighborhoods. However, their approach, albeit demonstrating a grouping effect (Li et al., 2020), restricts the optimization objective into a K-hop neighborhood, focusing on similar local structures and node features. Without explicit label specification, their methods struggle to ensure classification performance on diverse real-world graphs.

## 3 PRELIMINARIES

**Notations.** Let $\mathcal{G} = (\mathcal{V}, \mathcal{E})$ be a graph with node set $\mathcal{V}$ and edge set $\mathcal{E}$, where the number of nodes is denoted by $N$. We define the adjacency matrix as $\mathbf{A} \in \mathbb{R}^{N \times N}$ with $A_{i,j}$ refers to the weight of edge between node pairs $v_i, v_j \in \mathcal{V}$. The degree matrix $\mathbf{D}$ can be obtained by summing the rows of $\mathbf{A}$ into a diagonal matrix. We denote the graph Laplacian matrix as $\mathbf{L} = \mathbf{D} - \mathbf{A}$, which is often normalized into $\hat{\mathbf{L}} = \mathbf{I}_N - \hat{\mathbf{A}}$ with $\hat{\mathbf{A}} = \mathbf{D}^{-\frac{1}{2}}\mathbf{A}\mathbf{D}^{-\frac{1}{2}}$ and $\mathbf{I}_N$ being an identity matrix. Let $\hat{\mathbf{L}} = \mathbf{U}\mathbf{\Lambda}\mathbf{U}^T$ be the spectral decomposition of $\hat{\mathbf{L}}$, where the columns of $\mathbf{U}$ refer to eigenvectors and $\mathbf{\Lambda} = \text{diag}(\lambda_1, \lambda_2, \cdots, \lambda_N)$ consists of eigenvalues with each $\lambda_n \in [0, 2]$. For node classification on graph $\mathcal{G}$, nodes are associated with a feature matrix $\mathbf{X} \in \mathbb{R}^{N \times f}$ with $f$ being raw feature dimensions, and each of them is assigned a class $c_i$ with a one-hot vector $\mathbf{y}_i \in \mathbb{R}^C$ where $C \leq N$ is class number.

**Spectral Filtering.** Spectral filtering is essential in spectral GNNs. It selectively shrinks or amplifies the Fourier coefficients of node features (Defferrard et al., 2016) and usually take the form as

$$\mathbf{Z} = g_\psi(\hat{\mathbf{L}})\mathbf{X} = \mathbf{U}g_\psi(\mathbf{\Lambda})\mathbf{U}^T\mathbf{X}. \tag{1}$$

Here, $g_\psi : [0, 2] \to \mathbb{R}$ defines a graph filter function, which are often approximated by a $K$-order polynomial in practice. Specifically, we have $g_\psi(\lambda) = \sum_{k=0}^{K} \psi_k P_k(\lambda) = \sum_{k=0}^{K} \omega_k \lambda^k$ where $P_k : [0, 2] \to \mathbb{R}$ refers to a polynomial basis and both $\psi_k$ and $\omega_k$ denote the polynomial coefficient.

**Spatial Aggregation.** Spatial Aggregation is a central component of spatial GNNs, facilitating the propagation of node information along graph edges and its subsequent aggregation within node neighborhood. To provide a more formal illustration of spatial aggregation, let's consider the widely adopted GNN model, APPNP (Gasteiger et al., 2019). This model begins by applying a feature transformation, given by $\mathbf{Z}^{(0)} = f(\mathbf{X})$. The propagation then proceeds as:

$$\mathbf{Z}^{(k)} = (1 - \alpha)\mathbf{Z}^{(0)} + \alpha\hat{\hat{\mathbf{A}}}\mathbf{Z}^{(k-1)}, \quad k = 1, 2, \cdots, K, \tag{2}$$

where $\hat{\hat{\mathbf{A}}} = \tilde{\mathbf{D}}^{-\frac{1}{2}}\tilde{\mathbf{A}}\tilde{\mathbf{D}}^{-\frac{1}{2}}$, $\tilde{\mathbf{A}} = \mathbf{A} + \mathbf{I}$, and $\alpha$ refers to the update rate.

# 4 RETHINKING SPECTRAL GNNS FROM THE SPATIAL PERSPECTIVE

In this section, we provide both theoretical and empirical analyses to examine spectral GNNs from the spatial perspective and answer the question, *i.e.*, what information is essentially encoded by spectral GNNs in the spatial domain?

## 4.1 INTERPLAY OF SPECTRAL AND SPATIAL DOMAINS THROUGH THE LENS OF GRAPH OPTIMIZATION

The graph signal denoising problem (Shuman et al., 2013) was initially leveraged in (Ma et al., 2021; Zhu et al., 2021) as a means to interpret GNNs with smoothness assumption, which yet does not always hold in certain real-world graph scenarios such as heterophily (Zhu et al., 2020a). Without loss of generality, in this paper, we consider a more generalized graph optimization problem [1]

$$\arg\min_{\mathbf{Z}} \ \mathcal{L} = \|\mathbf{X} - \mathbf{Z}\|_F^2 + c \cdot \mathrm{tr}(\mathbf{Z}^T \gamma_\theta(\hat{\mathbf{L}})\mathbf{Z}) \tag{3}$$

where $\mathbf{Z} \in \mathbb{R}^{N \times d}$ refers to node representations, $\gamma_\theta(\hat{\mathbf{L}})$ determines the rate of propagation (Spielman, 2012) by operating on the graph spectrum, i.e., $\gamma_\theta(\hat{\mathbf{L}}) = \mathbf{U}\gamma_\theta(\mathbf{\Lambda})\mathbf{U}^T$, and $c$ is a trade-off coefficient. In case of setting $\gamma_\theta(\hat{\mathbf{L}}) = \hat{\mathbf{L}}$, Eq. (3) turns into the well-known graph signal denoising problem. To ensure the convexity of the objective in Eq. (3), a positive semi-definite constraint is imposed on $\gamma_\theta(\hat{\mathbf{L}})$, i.e., $\gamma_\theta(\lambda) \geq 0$ for $\lambda \in [0, 2]$. Then, one can address this minimization problem through either closed-form or iterative solutions.

**Closed-form Solution.** The closed-form solution can be obtained by setting the derivative of the objective function $\mathcal{L}$ to 0, i.e., $\frac{\partial \mathcal{L}}{\partial \mathbf{Z}} = 2(\mathbf{Z} - \mathbf{X}) + 2c\gamma_\theta(\hat{\mathbf{L}})\mathbf{Z} = 0$. Let $g_\psi(\lambda) = (1 + c\gamma_\theta(\lambda))^{-1}$, we can observe that the closed-form solution in Eq. (4) is equivalent to the spectral filtering in Eq. (1).

$$\mathbf{Z}^* = (\mathbf{I} + c\gamma_\theta(\hat{\mathbf{L}}))^{-1}\mathbf{X} = \mathbf{U}(\mathbf{I} + c\gamma_\theta(\mathbf{\Lambda}))^{-1}\mathbf{U}^T\mathbf{X} = \mathbf{U}g_\psi(\mathbf{\Lambda})\mathbf{U}^T\mathbf{X}. \tag{4}$$

As $\gamma_\theta(\lambda) \geq 0$, this establishes a more stringent constraint for the graph filter in spectral GNNs, i.e., $0 < g_\psi(\lambda) \leq \frac{1}{(1+c\cdot0)} = 1$, which is termed as a non-negative constraint in this paper.

**Iterative Solution.** Alternatively, we can take an iterative gradient descent method with a step size $\frac{1}{2+2c}$, which yields a concise iterative solution in Eq. (5) with $\hat{\mathbf{A}}^{\mathrm{new}} = \mathbf{I} - \gamma_\theta(\hat{\mathbf{L}})$. Notably, by taking $\hat{\mathbf{A}}^{\mathrm{new}}$ as a new computation graph, this solution closely mirrors the spatial aggregation in Eq. (2).

$$\mathbf{Z}^{(k)} = \frac{1}{1+c}\mathbf{X} + \frac{c}{1+c}\hat{\mathbf{A}}^{\mathrm{new}}\mathbf{Z}^{(k-1)}, \quad k = 1, 2, \cdots, K \tag{5}$$

**Theoretical Interaction — the Adapted New Graph.** With the non-negative constraint, it is evident that both spectral filtering and spatial aggregation effectively address the generalized graph optimization problem in Eq. (3), despite their distinctive forms and operation domains. Upon closer examination, we discover a compelling relationship between the graph filter $g_\psi(\lambda)$ in Eq. (4) and the new graph $\hat{\mathbf{A}}^{\mathrm{new}}$ in Eq. (5), given $g_\psi(\lambda) = (1 + c\gamma_\theta(\lambda))^{-1}$, that is,

$$\hat{\mathbf{A}}^{\mathrm{new}} = \mathbf{I} - \gamma_\theta(\hat{\mathbf{L}}) = \mathbf{I} - c^{-1}(g_\psi(\hat{\mathbf{L}})^{-1} - \mathbf{I}) \tag{6}$$

which unveils an intrinsic inter-play, *i.e.*, spectral filtering implicitly leads the original graph to an adapted new graph, explicitly computed for spatial aggregation.

It is worthy noting that this new graph fundamentally differs from the graph operation $g_\psi(\hat{\mathbf{L}})$ used in spectral filtering. Whereas the former elucidates the inherent spatial node relationships, the latter primarily processes node features without a clear physical interpretation. Moreover, it is crucial to understand that $g_\psi(\hat{\mathbf{L}})$ may not be fully-connected, especially with fixed-order polynomial approximation. This is because it only captures up to a $K$-hop neighborhood, i.e., $g_\psi(\hat{\mathbf{L}}) = \sum_{k=0}^{K} \psi_k P_k(\hat{\mathbf{L}}) = \sum_{k=0}^{K} \omega_k \hat{\mathbf{A}}^k$. In contrast, this uncovered new graph intrinsically enjoys a non-local property, as confirmed in the following section. Building upon this inherent non-locality, we further devise a framework to overcome the limitations of the truncated polynomials (see details in Section 5).

---

[1]This problem was first introduced in (He et al., 2021) to derive theoretically grounded graph filters. However, in this study, we repurpose it as a bridge between spectral filtering and spatial aggregation.

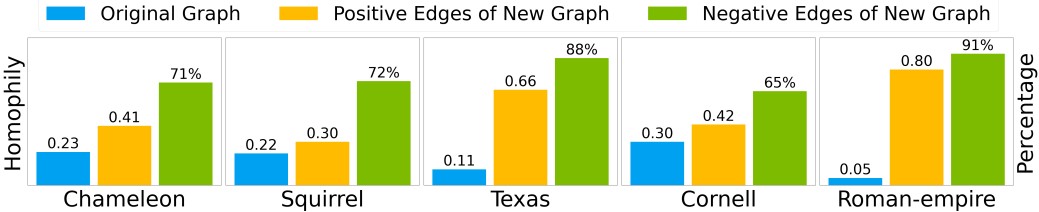

Figure 1: Distributions of adjacent nodes in the new graph based on their geodesic distance in the original graph. It is seen that nodes, distant in the original graph, can be connected in the new graph.

Figure 2: Left y-axis: Comparison of homophily between the original graph and the new graph (considering only positive edges). Right y-axis: the percentage (%) of label discrepancies as identified by negative edges. More details and visualizations can be found in Appendix F.1.

## 4.2 IN-DEPTH ANALYSIS OF THE ADAPTED NEW GRAPH

To deepen our understanding of the interpretability produced by spectral GNNs in the spatial domain, we embark upon a blend of theoretical and empirical inquiries into the adapted new graph.

### 4.2.1 NON-LOCALITY.

Our examination of the adapted new graph illuminates its non-local nature, particularly evident in the infinite series expansion of the original graph's adjacency matrix. To elucidate, we first introduce an pivotal mathematical construct, the Neumann series, in the following lemma.

**Lemma 1.** *Let $\mathbf{M} \in \mathbb{R}^{N \times N}$ be a matrix with eigenvalues $\lambda_n$, if $|\lambda_n| < 1$ for all $n = 1, 2, ..., N$, then $(\mathbf{I} - \mathbf{M})^{-1}$ exists and can be expanded as an infinite series, i.e., $(\mathbf{I} - \mathbf{M})^{-1} = \sum_{t=0}^{\infty} \mathbf{M}^t$, which is known as Neumann series.*

With the established non-negative constraint on graph filters, specifically $0 < g_\psi(\lambda) \leq 1$, it becomes evident that the eigenvalues of $\mathbf{I} - g_\psi(\hat{\mathbf{L}})$ falls into the interval permitting Neumann series expansion, as shown in lemma 1 (Horn & Johnson, 2012). Building on this observation, we present a non-trivial property of the new graph in the following proposition (see proof in Appendix B.1).

**Proposition 1.** *Given adjacency matrix $\hat{\mathbf{A}}^{new}$ formulated in Eq. (6), the adapted new graph exhibits non-locality. Specifically, $\hat{\mathbf{A}}^{new}$ is expressible as an infinite series expansion of the original graph's adjacency matrix $\hat{\mathbf{A}}$. Formally, we have $\hat{\mathbf{A}}^{new} = \mathbf{I} - c^{-1} \sum_{t=1}^{\infty} (\mathbf{I} - \sum_{k=0}^{K} \pi_k \hat{\mathbf{A}}^k)^t = \sum_{t=0}^{\infty} \phi_t \hat{\mathbf{A}}^t$ where $\pi_k$ and $\phi_t$ refer to the constant coefficients computed from $\{\psi_0, \psi_1, ..., \psi_K\}$ in distinct ways.*

This proposition implies that the new graph engenders immediate links between nodes that originally necessitate multiple hops for connection. To further underpin this theoretical claim, we analyze the general connection status on the new graph by BernNet (He et al., 2021), a spectral GNN adhering to the non-negative constraint. From Figure 1, it is apparent that nodes originally separated by multiple hops achieve direct connections in the new graph. See more empirical evidences in Appendix F.1.

### 4.2.2 SIGNED EDGE WEIGHTS — DISCERNING LABEL CONSISTENCY.

Upon further scrutinizing the adapted new graph, we make a notable discovery that it readily accommodates both positive and negative edge weights. A more granular analysis in Figure 2 reveals that a considerable portion of positive edge weights are assigned to the same-class node pairs, enhancing graph homophily (see the formula in Appendix D). Conversely, edges parameterized with negative

weights tend to bridge nodes with different labels. These findings demonstrate the newfound graph's adeptness in discerning label consistency among nodes. To theoretically explain this phenomenon, we further present the proposition below (see proof in Appendix B.2):

**Proposition 2.** *Let $\mathbf{Z}^*$ be the node representations optimized by Eq. (3). For $\mathbf{Z}^*$ to be effective in label prediction, it is a necessary condition that $\hat{\mathbf{A}}^{new}$ accommodates both positive and negative edge weights s.t. for any node pairs $v_i, v_j \in \mathcal{V}$, $\hat{A}_{i,j}^{new} > 0$ if $\mathbf{y}_i = \mathbf{y}_j$ and $\hat{A}_{i,j}^{new} < 0$ if $\mathbf{y}_i \neq \mathbf{y}_j$.*

Proposition 2 provides a theoretical foundation of our empirical findings on the new graph. The essence lies in the objective in Eq. (3), particularly the trace term $\mathrm{tr}(\mathbf{Z}^T \gamma_\theta(\hat{\mathbf{L}})\mathbf{Z})$. For clarity, let us reinterpret this trace term as $\mathrm{tr}(\bar{\mathbf{Z}}^T(\mathbf{D}^{new} - \mathbf{A}^{new})\bar{\mathbf{Z}})$, where $\mathbf{D}^{new}$ denotes the related degree matrix and $\bar{\mathbf{Z}}$ is derived from rescaling $\mathbf{Z}$. Clearly, this term evaluates label smoothness among adjacent nodes in the new graph, which, given its non-local nature, includes both intra-class ($=$) and inter-class ($\neq$) node connections such that $\mathbf{A}^{new} = \mathbf{A}^{new}_= + \mathbf{A}^{new}_{\neq}$. Drawing from proposition 2, we can further dissect the original trace term, splitting it into $\mathrm{tr}(\bar{\mathbf{Z}}^T(\mathbf{D}^{new}_= - \mathbf{A}^{new}_=)\bar{\mathbf{Z}}) - \mathrm{tr}(\bar{\mathbf{Z}}^T|(\mathbf{D}^{new}_{\neq} - \mathbf{A}^{new}_{\neq})|\bar{\mathbf{Z}})$ where the $|\cdot|$ operation denotes absolute values. As such, it becomes evident that minimizing this trace term not only enhances the representational proximity for same-class node pairs but also strengthens the distinctiveness for different-class nodes pairs. Such nuanced behaviors, inherent to the optimization in Eq. (3), are necessary for GNN models to achieve accurate label predictions.

To summarize, our investigation into spectral GNNs in the spatial domain reveals that graph spectral filtering fundamentally alters the original graph, imbuing it with non-locality and signed edge weights that capture label consistency among nodes. These findings highlight the interpretable role of spectral GNNs in the spatial domain, and prompt us to rethink current spectral GNNs beyond the truncated polynomial filters.

## 5 SPATIALLY ADAPTIVE FILTERING FRAMEWORK

Building on our discoveries, we re-evaluate the state-of-the-art spectral GNNs and put forth a paradigm-shifting framework, Spatially Adaptive Filtering (SAF), for joint exploitation of graph-structured data across both spectral and spatial domains. SAF leverages the adapted new graph by spectral filtering for an auxiliary non-local aggregation, addressing enduring challenges in GNNs related to long-range dependencies and graph heterophily. See the overall pipeline in Appendix A.

**Non-negative Spectral Filtering.** The proposed SAF requires explicit computation of the newfound graph, as outlined in Eq. (6). This further necessitates the graph filter $g_\psi : [0, 2] \to \mathbb{R}$ to satisfy the non-negative constraint from Eq. (3): $0 \leq g_\psi(\lambda) \leq 1$. However, not all extant graph filters fulfill this prerequisite. For instance, the filter use by GCN (Kipf & Welling, 2017), $g_\psi(\lambda) = 1 - \lambda$, takes negative values when $\lambda > 1$. In this research, we approximate the graph filter using Bernstein polynomials (Farouki, 2012), which are known for their non-negative traits (Powers & Reznick, 2000) and are essential in a preeminent spectral GNN, BernNet (He et al., 2021). For $g_\psi(\lambda) \leq 1$ part, we rescale Bernstein polynomials with the following proposition.

**Proposition 3.** *Let $B_{k,K}(x)$ denote the Bernstein polynomial basis of index $k$ and degree $K$, which is defined as $B_{k,K}(x) = \binom{K}{k}(1-x)^{K-k}x^k$ for $x \in [0, 1]$. Let $\psi_k$ denote the $k$-th coefficient of a polynomial $p(x)$ of degree $K$, where $p(x) = \sum_{k=0}^{K} \psi_k B_{k,K}(x)$ with $\psi_k \geq 0$ for all $k$. Then for all $x \in [0, 1]$, we have $g_\psi(x) \leq \max\{\psi_k\}_{k=0}^{K}$.*

Proposition 3 suggests that the Bernstein polynomial function attains its maximum value in $\psi_{\max} = \max\{\psi_k\}_{k=0}^{K}$. Therefore, $g_\psi(\lambda)$ can be rescaled within $[0, 1]$ by $\hat{g}_\psi(\lambda) = \frac{1}{\psi_{\max}} \sum_{k=0}^{K} \psi_k B_{k,K}(\frac{\lambda}{2})$, enabling us to formulate the spectral filtering in SAF as

$$\mathbf{Z}_f = \hat{g}_\psi(\hat{\mathbf{L}})f_\varphi(\mathbf{X}) = \frac{1}{\psi_{\max}} \sum_{k=0}^{K} \psi_k \frac{1}{2^K} \binom{K}{k} (2\mathbf{I} - \hat{\mathbf{L}})^{K-k} \hat{\mathbf{L}}^k f_\varphi(\mathbf{X})$$

where $f_\varphi(\cdot)$, a two-layer MLP, maps $\mathbf{X}$ from $f$ to $C$ dimensions using 64 hidden units, and $\{\psi_k\}_{k=1}^{K}$ are non-negative learnable parameters. See Appendix G for alternative spectral filters in SAF.

**Non-local Spatial Aggregation.** Once acquiring a suitable spectral filter $\hat{g}_\psi(\lambda)$, we promptly compute the adapted new graph using Eq. (6), represented as $\hat{\mathbf{A}}^{new} = \mathbf{I} - \tau(\mathbf{U}g_\psi(\mathbf{\Lambda})^{-1}\mathbf{U}^T - \mathbf{I})$ where

$\tau = c^{-1}$ serves as a scaling hyper-parameter and spectral decomposition is utilized for efficient matrix inversion. Equipped with this newfound graph, we proceed to perform non-local aggregation:

$$\mathbf{Z}^{(l)} = (1 - \eta)\mathbf{Z}^{(0)} + \eta\hat{\mathbf{A}}^{\text{new}}\mathbf{Z}^{(l-1)}, \quad l = 1, 2, \cdots, L,$$

where $\eta$ refers to the update rate and $\mathbf{Z}^{(0)} = f_\varphi(\mathbf{X})$. The iteratively aggregated results are denoted as $\mathbf{Z}_a$. Recognizing the potential noise from the non-local nature of $\hat{\mathbf{A}}^{\text{new}}$, we employ a sparsification technique, leveraging a positive threshold $\epsilon$, and retain only essential elements outside the $[-\epsilon, \epsilon]$ interval. For clarity, this refined model is referred to as SAF-$\epsilon$.

**Node-wise Prediction Amalgamation.** To optimally use information from different graph domains, our method employs an attention mechanism, allowing nodes to determine the importance of each space. Rather than merging features directly, which could obscure domain specifics, this mechanism produces pairwise weights for a nuanced amalgamation during model prediction. Specifically, the weight pair is computed as $\boldsymbol{\kappa}_f = \text{Sigmoid}(\mathcal{P}_f(\mathbf{Z}_f)), \boldsymbol{\kappa}_a = \text{Sigmoid}(\mathcal{P}_a(\mathbf{Z}_a))$ where $\mathcal{P}_f(\cdot)$ and $\mathcal{P}_a(\cdot)$ are two different mapping functions respectively for spectral and spatial features and $\boldsymbol{\kappa}_f, \boldsymbol{\kappa}_a \in \mathbb{R}^N$. For simplicity, we implement them using two one-layer MLPs. Given domain predictions $\mathbf{Y}_f = \text{Softmax}(\mathbf{Z}_f)$ and $\mathbf{Y}_a = \text{Softmax}(\mathbf{Z}_a)$, the final model prediction is attained as

$$\mathbf{Y} = \text{diag}(\boldsymbol{\kappa}_f) \cdot \mathbf{Y}_f + \text{diag}(\boldsymbol{\kappa}_a) \cdot \mathbf{Y}_a \text{ where } [\boldsymbol{\kappa}_f, \boldsymbol{\kappa}_a] \leftarrow \frac{[\boldsymbol{\kappa}_f, \boldsymbol{\kappa}_a]}{\max\{\|\{\boldsymbol{\kappa}_f, \boldsymbol{\kappa}_a\}\|_1, \delta\}}. \quad (7)$$

Here, $\delta$ is a small value to prevent non-zero division, with $\boldsymbol{\kappa}_f + \boldsymbol{\kappa}_a = \mathbf{1}$ maintained. Similar learning scheme can be founded in works (Sagi & Rokach, 2018; Wang et al., 2020; Zhu et al., 2020b).

**Complexity.** Compared to BernNet (He et al., 2021), SAF augments the design with spectral decomposition, non-local aggregation, and node-wise amalgamation. The spectral decomposition, precomputed at $\mathcal{O}(N^3)$ time and $\mathcal{O}(N^2)$ space complexities, is reusable for both training and inference (see more on its practical applications in Appendix C.3). Non-local aggregation involves creating a new graph and propagating information over it. In SAF-$\epsilon$, these two steps are distinctly separated by a sparsification procedure, culminating in a complexity of $\mathcal{O}(N^3 + N^2 + nnz(\hat{\mathbf{A}}^{\text{new}})d)$ where $nnz$ denotes non-zero element count. Conversely, the standard SAF, viewing non-local aggregation holistically, can reduce complexity to $\mathcal{O}(2dN^2 + dN)$ when $d \ll N$. For node-wise amalgamation, its parallelizable nature ensures computational efficiency. We present empirical studies on both time and space overheads in Appendix F.5.

## 6 EXPERIMENTS

**Datasets** We evaluate models over 13 real-world datasets from various domains. These include three well-known homophilic graphs: Cora, Citeseer, and Pubmed (Sen et al., 2008), five commonly used heterophilic graphs: Chameleon, Squirrel (Rozemberczki et al., 2021), Cornell, Texas (Pei et al., 2020), and Actor (Tang et al., 2009), as well as five recently introduced benchmarks: Twitch-DE (Lim et al., 2021) (social network), Minesweeper (synthetic graph), Tolokers (crowdsourcing platform worker network), Amazon-ratings (product co-purchasing), and Roman-empire (word dependency graph) (Platonov et al., 2023). We list their statistics in Appendix D.

**Baselines.** (1) MLP; (2) basic GNNs: GCN (Kipf & Welling, 2017) and APPNP (Gasteiger et al., 2019); (3) SOTA spatial GNNs: GCNII (Chen et al., 2020), FAGCN (Bo et al., 2021), GEN (Wang et al., 2021), PDE-GCN (Eliasof et al., 2021), NodeFormer (Wu et al., 2022) and GloGNN++ (Li et al., 2022); (4) SOTA spectral GNNs: ARMA (Bianchi et al., 2021), GPR-GNN (Chien et al., 2021), BernNet (He et al., 2021), ChebNetII (He et al., 2022), JacobiConv (Wang & Zhang, 2022), Specformer (Bo et al., 2023), LON-GNN (Tao et al., 2023) and OptBasisGNN (Guo & Wei, 2023a); (5) Unified GNNs: GNN-LF (Zhu et al., 2021), GNN-HF (Zhu et al., 2021), ADA-UGNN (Ma et al., 2021) and FE-GNN (Sun et al., 2023). Models like Geom-GCN (Pei et al., 2020) and Non-Local GNNs (Liu et al., 2021), surpassed by these SOTA methods, are excluded from our comparison.

**Setup.** To follow (He et al., 2021; 2022; Wang & Zhang, 2022), we fix $K = 10$. For each dataset, we perform a grid search to tune the hyper-parameters of all models. With the best hyper-parameters, we train models with Adam optimizer (Kingma & Ba, 2014) in 1,000 epochs using early-stopping strategy and a patience of 200 epochs, and report the mean classification accuracies with a 95% confidence interval on 10 random data splits. More experimental details can be found in Appendix E.

Table 1: Semi-supervised node classification accuracy (%) $\pm$ 95% confidence interval.

| Method | Cham. | Squi. | Texas | Corn. | Actor | Cora | Cite. | Pubm. |
|---|---|---|---|---|---|---|---|---|
| MLP | 26.36±2.85 | 21.42±1.50 | 32.42±9.91 | 36.53±7.92 | 29.75±0.95 | 57.17±1.34 | 56.75±1.55 | 70.52±2.01 |
| GCN | 38.15±3.77 | 31.18±0.93 | 34.68±9.07 | 32.36±8.55 | 22.74±2.37 | 79.19±1.37 | 69.71±1.32 | 78.81±0.84 |
| APPNP | 32.73±2.31 | 24.50±0.89 | 34.79±10.11 | 34.85±9.71 | 29.74±1.04 | 82.39±0.68 | 69.79±0.92 | 79.97±1.58 |
| ARMA | 37.42±1.72 | 24.15±0.93 | 39.65±8.09 | 28.90±10.07 | 27.02±2.31 | 79.14±1.07 | 69.35±1.44 | 78.31±1.33 |
| GPR-GNN | 33.03±1.92 | 24.36±1.52 | 33.98±11.90 | 38.95±12.36 | 28.58±1.01 | 82.37±0.91 | 69.22±1.27 | 79.28±2.25 |
| ChebNetII | **43.42±3.54** | **33.96±1.22** | 46.58±7.68 | 42.19±11.61 | 30.18±0.81 | 82.42±0.64 | 69.89±1.21 | 79.51±1.03 |
| JacobiConv | 36.67±1.63 | 29.38±0.71 | 48.50±5.90 | 43.01±11.92 | 31.69±0.71 | 82.93±0.55 | 70.25±1.02 | 79.53±1.28 |
| Specformer | 36.05±3.47 | 29.64±0.88 | 50.00±8.33 | 43.76±5.84 | 31.45±0.68 | 81.44±0.63 | 66.11±0.98 | 78.05±1.03 |
| LON-GNN | 35.17±1.85 | 30.25±1.04 | 45.38±7.92 | 35.32±8.09 | 31.51±1.23 | 81.93±0.74 | 70.41±1.10 | 79.57±1.08 |
| OptBasisGNN | 35.56±2.86 | 31.25±1.06 | 37.11±5.09 | 32.31±7.11 | 31.73±0.50 | 78.69±0.86 | 63.46±1.30 | 77.38±0.98 |
| GNN-LF | 26.49±2.00 | 22.01±1.04 | 39.02±6.24 | 36.65±9.60 | 28.28±0.71 | 81.96±0.92 | 69.80±1.36 | 79.50±1.28 |
| GNN-HF | 35.57±2.26 | 22.36±1.26 | 44.80±5.67 | 38.79±11.62 | 29.15±0.78 | 81.15±0.78 | 69.68±0.73 | 79.10±1.19 |
| ADA-UGNN | 39.39±2.02 | 25.65±0.49 | 47.86±6.65 | 42.89±8.09 | 30.78±1.00 | 82.52±1.04 | 70.18±1.40 | 79.78±1.32 |
| FE-GNN | 38.23±1.66 | 31.67±1.60 | 47.40±5.90 | 41.21±8.96 | 26.20±0.76 | 77.00±0.74 | 61.24±1.26 | 75.63±1.33 |
| BernNet | 27.32±4.04 | 22.37±0.98 | 43.01±7.45 | 39.42±9.59 | 29.87±0.78 | 82.17±0.86 | 69.44±0.97 | 79.48±1.47 |
| SAF | 41.82±1.74 | 31.77±0.69 | 58.04±3.76 | 52.49±8.56 | 33.50±0.55 | 83.57±0.66 | 71.07±1.08 | 79.51±1.12 |
| SAF-$\epsilon$ | 41.88±2.04 | 32.05±0.40 | **58.38±3.47** | **53.41±5.55** | **33.84±0.58** | **83.79±0.71** | **71.30±0.93** | **80.16±1.25** |
| Improv. | 14.56% | 9.68% | 15.37% | 13.99% | 3.97% | 1.62% | 1.86% | 0.68% |

**Semi-supervised Node Classification.** In this task, we follow the experimental protocol established by (He et al., 2022) and compare our models with MLP, two basic GNNs, seven popular polynomial spectral GNNs, and four unified GNNs. For data splitting on homophilic graphs (Cora, Citeseer, and Pubmed), we apply the standard division (Yang et al., 2016) with 20 nodes per class for training, 500 nodes for validation, and 1,000 nodes for testing. On the other five heterophilic graphs, we leverage the sparse splitting (Chien et al., 2021) with 2.5%/2.5%/95% samples respectively for training/validation/testing. The results are reported in Table 1, where the best results are bold, the underlined letters denote the second highest accuracy, and the "Improv." column indicates the relative enhancement of our models compared to BernNet. We first observe that both SAF and SAF-$\epsilon$ substantially boosts its base model, BernNet, with gains reaching a notable 15.37%. This impressive enhancement is credited to their capacity to effectively exploit the task-beneficial information, which is implicitly encoded by spectral filtering in the spatial domain. This ability is particularly advantageous in contexts with limited supervision, where it allows effective leveraging of extra prior knowledge during training. Generally, our models outperforms competitors on all datasets except for Chameleon and Squirrel, where it still make considerable improvements on BernNet by 14.56% and 9.68% and maintains a second-place rank. Moreover, we can observe that SAF-$\epsilon$ generally delivers better results than SAF. This advantage arises from its thresholding sparsity, reducing noise in non-local graphs for more efficient graph learning. However, this enhancement also incurs higher computational costs, as illustrated in both Section 5 and Appendix F.5.

**Full-supervised Node Classification.** To bolster our evaluation for the full-supervised node classification task, we augment the baseline models (previously used in semi-supervised node classification) with seven cutting-edge spatial GNNs: GCNII, FAGCN, GEN, PDE-GCN, Node-Former, and GloGNN++. For all datasets, we randomly divide them into 60%/20%/20% for training/validation/testing by following (He et al., 2021; 2022). Table 2 summarizes the mean classification accuracies. Notably, provided with more training samples, our proposed methods beat the leading spectral and spatial GNNs, except on Squi. dataset where they achieve comparable results to Specformer. This exceptional performance stems from its non-local aggregation, with signed edge weights that globally address both node similarity and dissimilarity. This allows our models to effectively mitigate long-standing issues inherent in GNNs, such as long-range dependencies and graph heterophily. In addition to the eight common datasets for node classification, we also incorporated experimental results from five recently proposed benchmarks tailored for graph heterophily. Due to space limit, the detailed results of these additional benchmarks are presented in Appendix F.2.

**Ablation Study.** Despite the critical contribution of non-local aggregation to SAF, spectral filtering remains vital for discriminative node representation learning. Specifically, the quality of the adapted new graph fundamentally hinges on the graph spectral filters' training, as underscored by their theoretical interaction in Eq. (6). To provide empirical evidences, we conduct an ablation study on SAF by removing spectral filtering (i.e., $\kappa_f = 0, \kappa_a = 1$ in Eq. (7)). From Figure 3a, the absence of spectral filtering results in a noticeable accuracy drop, confirming the importance of spectral filtering in SAF. For a comprehensive ablation analysis of other SAF modules, refer to Appendix F.3.

Table 2: Full-supervised node classification accuracy (%) $\pm$ 95% confidence interval.

| Method | Cham. | Squi. | Texas | Corn. | Actor | Cora | Cite. | Pubm. |
|---|---|---|---|---|---|---|---|---|
| MLP | 46.59±1.84 | 31.01±1.18 | 86.81±2.24 | 84.15±3.05 | 40.18±0.55 | 76.89±0.97 | 76.52±0.89 | 86.14±0.25 |
| GCN | 60.81±2.95 | 45.87±0.88 | 76.97±3.97 | 65.78±4.16 | 33.26±1.15 | 87.18±1.12 | 79.85±0.78 | 86.79±0.31 |
| APPNP | 52.15±1.79 | 35.71±0.78 | 90.64±1.70 | 91.52±1.81 | 39.76±0.49 | 88.16±0.74 | 80.47±0.73 | 88.13±0.33 |
| ARMA | 60.21±1.00 | 36.27±0.62 | 83.97±3.77 | 85.62±2.13 | 37.67±0.54 | 87.13±0.80 | 80.04±0.55 | 86.93±0.24 |
| GPR-GNN | 67.49±1.38 | 50.43±1.89 | 92.91±1.32 | 91.57±1.96 | 39.91±0.62 | 88.54±0.67 | 80.13±0.84 | 88.46±0.31 |
| ChebNetII | 71.37±1.01 | 57.72±0.59 | 93.28±1.47 | 92.30±1.48 | 41.75±1.07 | 88.71±0.93 | 80.53±0.79 | 88.93±0.29 |
| JacobiConv | 74.20±1.03 | 57.38±1.25 | 93.44±2.13 | 92.95±2.46 | 41.17±0.64 | 88.98±0.46 | 80.78±0.79 | 89.62±0.41 |
| Specformer | 75.06±1.10 | **65.05±0.96** | 90.33±3.12 | 90.00±2.79 | 42.55±0.67 | 88.85±0.46 | 80.68±0.90 | 91.25±0.31 |
| LON-GNN | 73.00±2.20 | 60.61±1.69 | 87.54±3.45 | 84.47±3.45 | 39.10±1.59 | 89.44±1.12 | 81.41±1.15 | 90.98±0.64 |
| OptBasisGNN | 74.26±0.74 | 63.62±0.76 | 91.15±1.97 | 89.84±2.46 | 42.39±0.52 | 87.96±0.71 | 80.58±0.82 | 90.30±0.19 |
| GCNII | 63.44±0.85 | 41.96±1.02 | 80.46±5.91 | 84.26±2.13 | 36.89±0.95 | 88.46±0.82 | 79.97±0.65 | 89.94±0.31 |
| FAGCN | 69.30±1.53 | 50.13±2.53 | 91.64±2.30 | 89.51±2.79 | 41.56±0.97 | 88.60±0.99 | 80.12±0.96 | 89.78±0.35 |
| GEN | 68.82±0.96 | 56.05±1.04 | 92.30±2.30 | 90.49±1.80 | 41.08±2.08 | 89.21±0.54 | 79.40±0.59 | 90.40±0.24 |
| PDE-GCN | 66.01±1.56 | 48.73±1.06 | 93.24±2.03 | 89.73±1.35 | 39.76±0.74 | 88.62±1.03 | 79.98±0.97 | 89.92±0.38 |
| NodeFormer | 53.02±1.58 | 34.25±1.96 | 87.71±2.13 | 90.00±3.45 | 41.74±0.61 | 86.93±1.22 | 79.58±0.85 | 91.27±0.39 |
| GloGNN++ | 72.36±0.85 | 60.60±1.04 | 91.48±1.48 | 89.84±3.62 | 41.87±1.02 | 87.21±0.59 | 79.89±0.61 | 86.89±0.33 |
| GNN-LF | 53.74±1.29 | 36.15±0.86 | 76.07±2.62 | 78.36±2.46 | 38.39±0.81 | 88.51±0.89 | 79.84±0.56 | 89.86±0.23 |
| GNN-HF | 55.97±1.05 | 35.29±0.72 | 81.15±2.62 | 85.41±3.12 | 38.96±0.77 | 88.28±0.64 | 80.04±0.93 | 90.35±0.30 |
| ADA-UGNN | 61.09±1.51 | 42.02±1.26 | 84.92±3.12 | 83.61±3.44 | 41.10±0.62 | 88.74±0.85 | 79.81±1.11 | 90.61±0.44 |
| FE-GNN | 73.00±1.31 | 63.28±0.81 | 88.03±1.80 | 86.07±3.12 | 41.74±0.67 | 89.21±0.71 | 80.26±1.06 | 90.80±0.30 |
| BernNet | 68.53±1.68 | 51.39±0.92 | 92.62±1.37 | 92.13±1.64 | 41.71±1.12 | 88.51±0.92 | 80.08±0.75 | 88.51±0.39 |
| SAF | **75.30±0.96** | 63.63±0.81 | 94.10±1.48 | 92.95±1.97 | 42.93±0.79 | 89.80±0.69 | 80.61±0.81 | 91.49±0.29 |
| SAF-$\epsilon$ | 74.84±0.99 | 64.00±0.83 | **94.75±1.64** | **93.28±1.80** | **42.98±0.61** | **89.87±0.51** | **81.45±0.59** | **91.52±0.30** |
| Improv. | 6.77% | 12.61% | 2.13% | 1.15% | 1.27% | 1.36% | 1.37% | 3.01% |

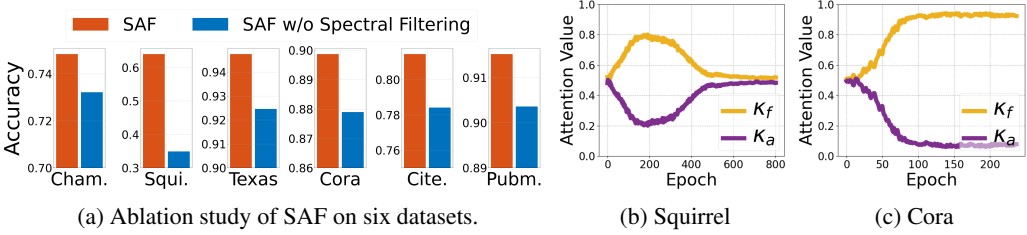

(a) Ablation study of SAF on six datasets.     (b) Squirrel     (c) Cora

Figure 3: (a) Comparison between SAF and its ablated variant in full-supervised node classification accuracy. (b-c) Changing trends of the averaged attention values with respect to training epochs.

**Analysis of Attention Trends.** We analyze the changing trends of the pair-wise attention weights during training SAF on Squirrel and Cora datasets. From Figure 9, the average weights for filtering and aggregation start similarly but diverge throughout training, showing different trends in heterophilic and homophilic graphs. On the heterophilic graph Squirrel, both weights converge to similar values, demonstrating their mutual importance in modeling complex connectivity. Conversely, $\kappa_f$ becomes dominant on the homophilic graph Cora due to the sufficiency of node proximity information for label prediction, thereby diminishing the relevance of $\kappa_a$ and non-local aggregation.

## 7 CONCLUSION AND FUTURE WORK

This paper introduces a fresh spatial perspective on spectral GNNs, shedding light on their interpretability. We reveal that spectral GNNs fundamentally leads the original graph to an adapted new one, which exhibits non-locality and accommodates signed edge weights to reflect label consistency among nodes. This insight leads to our proposed Spatially Adaptive Filtering (SAF) framework, enhancing spectral GNNs for more effective and versatile graph representation learning.

While SAF adeptly captures long-range dependencies and addresses graph heterophily, it has limitations. Specifically, its non-negative constraints on graph filters might limit filter expressiveness, indicating room for theoretical refinement. Furthermore, although this study focuses on a node-level investigation, it raises intriguing questions about the implications of spectral GNNs at the graph-level in the spatial domain. Future work could expand this examination by exploring the interplay between spatial and spectral domains from a more comprehensive graph-level viewpoint.

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

# A OVERALL PIPELINE OF SPATIALLY ADAPTIVE FILTERING FRAMEWORK

Figure 4: Illustration of the proposed SAF framework, where varying node colors on the input graph represent different node labels.

# B PROOFS

## B.1 PROOF OF PROPOSITION 1

*Proof.* We begin with the assertion that the eigenvalues of $\mathbf{I} - g_\psi(\hat{\mathbf{L}})$ are positive and strictly less than 1, which fulfills the necessary condition for the Neumann series expansion stated in Lemma 1. As such, we can deduce $g_\psi(\hat{\mathbf{L}})^{-1} = (\mathbf{I} - (\mathbf{I} - g_\psi(\hat{\mathbf{L}})))^{-1} = \sum_{t=0}^{\infty}(\mathbf{I} - g_\psi(\hat{\mathbf{L}}))^t$. Owning to the prevalent polynomial approximation, we are eligible to express $g_\psi(\hat{\mathbf{L}})$ w.r.t. adjacency matrix $\hat{\mathbf{A}}$, i.e., $g_\psi(\hat{\mathbf{L}}) = g_\psi(\mathbf{I} - \hat{\mathbf{A}}) = \sum_{k=0}^{K}\pi_k\hat{\mathbf{A}}^k$ where $\pi_k$ refers to the new coefficients made of up $\{\psi_m\}_{m=0}^{K}$. Substituting this polynomial representation into our Neumann expansion, we obtain $g_\psi(\hat{\mathbf{L}})^{-1} = \sum_{n=0}^{\infty}(\mathbf{I} - \sum_{k=0}^{K}\pi_k\hat{\mathbf{A}}^k)^t$. Now, revisiting $\hat{\mathbf{A}}^{\text{new}}$ formulated in Eq. (6), we have $\hat{\mathbf{A}}^{\text{new}} = \mathbf{I} - c^{-1}(g_\psi(\hat{\mathbf{L}})^{-1} - \mathbf{I}) = \mathbf{I} - c^{-1}\sum_{t=1}^{\infty}(\mathbf{I} - \sum_{k=0}^{K}\pi_k\hat{\mathbf{A}}^k)^t = \sum_{t=0}^{\infty}\phi_t\hat{\mathbf{A}}^t$ where $\phi_t$ denotes a constant coefficient made up of $\{\pi_m\}_{m=0}^{K}$. □

## B.2 PROOF OF PROPOSITION 2

*Proof.* Let us commence the proof by contradiction. Let $\mathcal{C}$ denote the condition described in proposition 2. Assume, for the sake of contradiction, that $\mathcal{C}$ is not requisite for the optimal node representations $\mathbf{Z}^*$ to be predictive of node labels. Under this assumption, there are node pairs $v_i, v_j \in \mathcal{V}$ such that: (1) if $\mathbf{y}_i = \mathbf{y}_j$, $\hat{A}_{i,j}^{\text{new}} < 0$; (2) if $\mathbf{y}_i \neq \mathbf{y}_j$, $\hat{A}_{i,j}^{\text{new}} > 0$. Without loss of generality, given the non-locality as proved in proposition 1, we exclude cases where $\hat{A}_{i,j}^{\text{new}} = 0$ from our consideration. Now, consider the second objective term $\text{tr}(\mathbf{Z}^T\gamma_\theta(\hat{\mathbf{L}})\mathbf{Z})$ in Eq. (3). Using the relationship $\gamma_\theta(\hat{\mathbf{L}}) = \mathbf{I} - \hat{\mathbf{A}}^{\text{new}}$, we can expand this term into $\sum_{v_i,v_j \in \mathcal{V}} \hat{A}_{i,j}^{\text{new}}\|\mathbf{Z}_i - \mathbf{Z}_j\|_2^2$. Under (1), for same-class nodes $v_i, v_j$ with $\hat{A}_{i,j}^{\text{new}} < 0$, minimizing the objective term pulls $\mathbf{Z}_i$ and $\mathbf{Z}_j$ apart in the representation space. This behavior violates the canonical understanding that nodes from the same class should exhibit similar representations. Under (2), for different-class nodes $v_i, v_j$ with $\hat{A}_{i,j}^{\text{new}} > 0$, the optimization encourages $\mathbf{Z}_i$ and $\mathbf{Z}_j$ to be more similar. This is in direct opposition to the basic classification principle that nodes from different classes should have distinct representations. Given these contradictions stemming from the mathematical implications in optimization, we must reject assumptions (1) and (2), affirming the necessary condition $\mathcal{C}$ for accurate label prediction by $\mathbf{Z}^*$. □

### B.3 PROOF OF PROPOSITION 3

*Proof.* We denote $p(x) = \sum_{k=0}^{K} \psi_k \binom{K}{k}(1-x)^{K-k}x^k$ as a Bernstein polynomial with $\psi_k \geq 0$ for all $k$ and $\psi_{\max} = \max\{\psi_k\}_{k=0}^{K}$. Given $x \in [0,1]$, we can derive the following inequality as

$$p(x) = \sum_{k=0}^{K} \psi_k \binom{K}{k}(1-x)^{K-k}x^k \leq \psi_{\max} \sum_{k=0}^{K} \binom{K}{k}(1-x)^{K-k}x^k = \psi_{\max}(1-x+x)^K = \psi_{\max}.$$

Therefore, we have $p(x) \leq \max\{\psi_k\}_{k=0}^{K}$ for all $x \in [0,1]$. $\qquad\square$

## C FURTHER REMARKS ABOUT RELATED WORK

### C.1 POLYNOMIAL SPECTRAL GRAPH NEURAL NETWORKS

**Vanilla GCN (Kipf & Welling, 2017).** The vanilla GCN truncates Chebyshev polynomials to a simple first-order for efficient graph convolution, which functions as a low-pass filter, i.e., $\mathbf{Z} = w(\mathbf{I} + \mathbf{D}^{-\frac{1}{2}}\mathbf{A}\mathbf{D}^{-\frac{1}{2}})\mathbf{X}$. To ensure numerical stability, it further leverages a renormalization trick to replace $\mathbf{I} + \mathbf{D}^{-\frac{1}{2}}\mathbf{A}\mathbf{D}^{-\frac{1}{2}}$ with $(\mathbf{D} + \mathbf{I})^{-\frac{1}{2}}(\mathbf{A} + \mathbf{I})(\mathbf{D} + \mathbf{I})^{-\frac{1}{2}}$ and derives each graph convolution layer as

$$\mathbf{Z}^{(k+1)} = \tilde{\mathbf{D}}^{-\frac{1}{2}}\tilde{\mathbf{A}}\tilde{\mathbf{D}}^{-\frac{1}{2}}\mathbf{Z}^{(k)}\mathbf{W}^{(k)},$$

where $\tilde{\mathbf{A}} = \mathbf{A} + \mathbf{I}$, $\tilde{\mathbf{D}} = \mathbf{D} + \mathbf{I}$, $\mathbf{Z}^{(0)} = \mathbf{X}$, and $\mathbf{W}^{(k)}$ denotes learnable weights in the $k$-th layer.

**GPR-GNN (Chien et al., 2021).** GPR-GNN leverages the Generalized PageRank to approximate spectral graph filters with Monomial polynomial basis. The model architecture is formulated as

$$\mathbf{Z} = \sum_{k=0}^{K} \psi_k \hat{\tilde{\mathbf{A}}}^k f_\varphi(\mathbf{X}),$$

where $\hat{\tilde{\mathbf{A}}} = \tilde{\mathbf{D}}^{\frac{1}{2}}\tilde{\mathbf{A}}\tilde{\mathbf{D}}^{\frac{1}{2}}$, and $f_\varphi(\mathbf{X})$ refers to a linear map with parameters $\varphi$. The approximated graph filter is $g_\psi(\tilde{\lambda}) = \sum_{k=0}^{K} \psi_k \tilde{\lambda}^k$ with $\tilde{\lambda}$ being the eigenvalues of $\hat{\tilde{\mathbf{A}}}$.

**BernNet (He et al., 2021).** BernNet approximates graph filters with Bernstein polynomials to attain non-negative property, yielding the model expression as

$$\mathbf{Z} = \sum_{k=0}^{K} \psi_k \frac{1}{2^K}\binom{K}{k}(2\mathbf{I} - \hat{\mathbf{L}})^{K-k}\hat{\mathbf{L}}^k f_\varphi(\mathbf{X}).$$

The graph filter is defined as $g_\psi(\lambda) = \sum_{k=0}^{K} \psi_k \frac{1}{2^K}\binom{K}{k}(2-\lambda)^{K-k}\lambda^k$ and $\lambda$ is the eigenvalue of $\hat{\mathbf{L}}$.

**ChebNetII (He et al., 2022).** ChebNetII leverages Chebyshev interpolation to approximate graph filter while reducing the Runge phenomenon. This results in the following model structure:

$$\mathbf{Z} = \mathbf{Y} = \frac{2}{K+1}\sum_{k=0}^{K}\sum_{j=0}^{K} \gamma_j T_k(x_j) T_k(\hat{\mathbf{L}}) f_\varphi(\mathbf{X}).$$

Here, $T_k(x) = 2xT_{k-1}(x) - T_{k-2}(x)$ denotes the Chebyshev basis with $T_0(x) = 1$ and $T_1(x) = x$, and $x_j = \cos(\frac{(j+1/2)\pi}{K+1})$ refers to the Chebyshev nodes of $T_{K+1}(x)(\cdot)$. The graph filter corresponds to $g_\psi(\lambda) = \frac{2}{K+1}\sum_{k=0}^{K} \psi_k T_k(\lambda)$ with $\psi_k = \sum_{j=0}^{K} \gamma_j T_k(x_j)$.

**JacobiConv (Wang & Zhang, 2022).** JacobiConv approximates graph filters using orthogonal Jacobi polynomial basis and feature-wise filter learning. It can be expressed as

$$\mathbf{Z}_{[:,l]} = \sum_{k=0}^{K} \psi_{k,l} \sum_{s=0}^{k} \beta_{k,s}(2\mathbf{I} - \hat{\mathbf{L}})^s(-\hat{\mathbf{L}})^{k-s}\mathbf{X}\mathbf{W}_{[:,l]},$$

where $\beta_{k,s} = \frac{(k+a)!(k+b)!}{2^k s!(k+a-s)!(b+s)!(k-s)!}$ is a constant coefficient with hyper-parameters $a, b$ and $l$ denotes the $l$-th feature channel. Thus, the feature-wise graph filter is given as $g_{\psi,l}(\lambda) = \sum_{k=0}^{K} \psi_{k,l} \sum_{s=0}^{k} \beta_{k,s}(2-\lambda)^s(-\lambda)^{k-s}$.

Table 3: Statistics of real-world datasets. Diameter refers to the longest geodesic distance between nodes on the graph. Both $\mathcal{H}$, $\mathcal{H}_{\text{class}}$, and $\mathcal{H}_{\text{adjusted}}$ measure graph homophily ratio from 0 to 1.

| Dataset | # Nodes | # Edges | # Features | # Classes | Diameter | $\mathcal{H}$ | $\mathcal{H}_{\text{class}}$ | $\mathcal{H}_{\text{adjusted}}$ |
|---|---|---|---|---|---|---|---|---|
| **Chameleon** | 2,277 | 31,371 | 2,325 | 5 | 11 | 0.23 | 0.04 | 0.03 |
| **Squirrel** | 5,201 | 198,353 | 2,089 | 5 | 10 | 0.22 | 0.03 | 0.01 |
| **Actor** | 7,600 | 26,659 | 932 | 5 | 12 | 0.22 | 0.01 | 0.00 |
| **Texas** | 183 | 279 | 1,703 | 5 | 8 | 0.11 | 0.00 | -0.23 |
| **Cornell** | 183 | 277 | 1,703 | 5 | 8 | 0.30 | 0.02 | -0.08 |
| **Cora** | 2,708 | 5,278 | 1,433 | 7 | 19 | 0.81 | 0.77 | 0.77 |
| **Citeseer** | 3,327 | 4,552 | 3,703 | 6 | 28 | 0.74 | 0.63 | 0.67 |
| **Pubmed** | 19,717 | 44,324 | 500 | 5 | 18 | 0.80 | 0.66 | 0.69 |
| **Minesweeper** | 10,000 | 39,402 | 7 | 2 | 99 | 0.68 | 0.01 | 0.01 |
| **Tolokers** | 11,758 | 519,000 | 10 | 2 | 11 | 0.59 | 0.18 | 0.09 |
| **Amazon-ratings** | 24,492 | 93,050 | 300 | 5 | 46 | 0.38 | 0.13 | 0.14 |
| **Roman-empire** | 22,662 | 32,927 | 300 | 18 | 6,824 | 0.05 | 0.02 | -0.05 |
| **Twtich-DE** | 9,498 | 153,138 | 2,514 | 2 | 7 | 0.63 | 0.14 | 0.14 |

Despite the proliferation of various filtering strategies, they invariably resort to approximating graph filters with fixed-order polynomials. While this approach does offer computational advantages, the truncated approximation limits the effective propagation range and hinder the ability to capture the long-range dependencies.

## C.2 INTERTWINED ISSUES OF LONG-RANGE DEPENDENCIES AND GRAPH HETEROPHILY

Long-range dependencies and graph heterophily are intrinsically connected. The necessity to capture long-range dependencies often arises in heterophilic (disassortative) graphs, while the phenomenon of graph heterophily further intensifies the challenges associated with long-range dependencies. In light of this, we aim to conduct a thorough evaluation of our model, comparing it to methods (except for polynomial spectral GNNs) that are designed with singular or dual focus on capturing long-range dependencies and addressing graph heterophily. From the plethora of models extant in the literature, we carefully select a number of representative and advanced methods for comparison. These include three methods for solely capturing long-range dependencies: APPNP (Gasteiger et al., 2019), GCNII (Chen et al., 2020), PDE-GCN (Eliasof et al., 2021), two models focusing on tackling graph heterophily: GEN (Wang et al., 2021), FAGCN (Bo et al., 2021), as well as two works considering both: NodeFormer (Wu et al., 2022), GloGNN++ (Li et al., 2022).

## C.3 PRACTICAL SIGNIFICANCE OF SPECTRAL DECOMPOSITION

Spectral decomposition, or eigendecomposition, breaks down a matrix into its eigenvalues and eigenvectors, offering insights into matrix properties, especially for the graph Laplacian. Even with its computational demands, this technique has attracted surging interest in the graph learning community due to its theoretical richness and can be practically expedited for larger graphs using Sparse Generalized Eigenvalue algorithms (Cai et al., 2021). Recent advancements underscore its value in various applications such as graph positional encoding (Belkin & Niyogi, 2003; Wang et al., 2022; Dwivedi et al., 2020; Lim et al., 2022), spectral graph convolution (Bo et al., 2023; Liao et al., 2019), graph domain adaptation (You et al., 2022), and graph robustness (Chang et al., 2021). For example, Laplacian eigenvectors have be widely used in identifying nodes' global position in the graph (Wang et al., 2022), particularly in recent popular graph transformers (Kreuzer et al., 2021; Kim et al., 2022; Rampášek et al., 2022), enhancing their expressiveness. Innovations like SignNet and BasisNet (Lim et al., 2022) have further optimized the processing of these eigenvectors. When exploring the expressive power of GNN models, Specformer (Bo et al., 2023) employs spectral decomposition for learning set-to-set graph filters, while FE-GNN (Sun et al., 2023) taps into singular value decomposition (SVD) for graph feature expansion. In line with these developments, our method, SAF, utilizes spectral decomposition to explicitly create a new graph, enabling efficient non-local aggregation to capture long-range dependencies and address graph heterophily.

## D    DATASET INFORMATION

We conduct experiments on 13 real-world datasets from various domains. The detailed statistics are summarized in Table 3. Alongside common data attributes, we also provide the longest geodesic (shortest-path) distance between graph nodes for better illustrating the non-local property we investigate in Figure 1 and Figure 5. Moreover, we employ three metrics - edge homophily (Zhu et al., 2020a) $\mathcal{H}$, class homophily (Lim et al., 2021) $\mathcal{H}_{\text{class}}$, and adjusted homophily (Platonov et al., 2023) $\mathcal{H}_{\text{adjusted}}$ - to assess the graph's homophily ratio, which ranges from 0 (high heterophily) to 1 (high homophily). While the first is a commonly used index, the latter two, considering class variability and potential imbalance, have been recently introduced to provide a more accurate estimation. For our main text analysis regarding the adapted new graph, we primarily rely on the edge homophily metric, defined as $\mathcal{H} = |\{(v_i, v_j)|(v_i, v_j) \in \mathcal{E} \wedge \mathbf{y}_i = \mathbf{y}_j\}|/|\mathcal{E}|$, given its simplicity and wide usage. In certain compact sections of this paper, we use four-letter abbreviations for dataset names. As these datasets are open benchmarks, we have downloaded and utilized them from the respective web links provided below.

- `https://github.com/ivam-he/ChebNetII/tree/main/main/data`
- `https://github.com/yandex-research/heterophilous-graphs/tree/main/data`
- `https://github.com/CUAI/Non-Homophily-Benchmarks/tree/main/data/twitch`

## E    EXPERIMENTAL DETAILS

In this section, we provide experimental details for reproducibility. As He et al. (2022) have made a comprehensive evaluation and share the same experimental protocol with us, we directly leverage their results for models: MLP, GCN, ARMA, APPNP, GCNII, PDE-GCN, GPR-GNN, BernNet, and ChebNetII on datasets including Chameleon, Squirrel, Texas, Cornell, Actor, Cora, Citeseer, and Pubmed. Other experiments are performed on a machine equipped with an NVIDIA GeForce RTX 3090 (24GB) and an Intel(R) Xeon(R) Gold 5218R CPU @ 2.10GHz (20 cores).

### E.1    OUR IMPLEMENTATIONS

The main codes for our proposed SAF can be founded in the supplementary material.

### E.2    BASELINE IMPLEMENTATIONS

In our experiments, we leverage the Pytorch Geometric library[2] implementations for GCN and APPNP. For MLP, we include a sequence of linear layers, each of which is followed by batch normalization, ReLU activation, and dropout. The number of MLP layers are tuned from 1 to 5. For the remaining baselines, we resort to the officially released code, accessible via the provided URLs. Besides, for the work (Li et al., 2022), only the most effective model variant, GloGNN++, is included in our experiments.

- GPR-GNN: `https://github.com/jianhao2016/GPRGNN`
- BernNet: `https://github.com/ivam-he/BernNet`
- ChebNetII: `https://github.com/ivam-he/ChebNetII`
- JacobiConv: `https://github.com/GraphPKU/JacobiConv`
- FAGCN: `https://github.com/bdy9527/FAGCN`
- GEN: `https://github.com/BUPT-GAMMA/Graph-Structure-Estimation-Neural-Networks`
- NoderFormer: `https://github.com/qitianwu/NodeFormer`
- GloGNN++: `https://github.com/RecklessRonan/GloGNN`
- Specformer: `https://github.com/DSL-Lab/Specformer`

---

[2]`https://github.com/pyg-team/pytorch_geometric`

- **LON-GNN**: `https://github.com/TaoLbr1993/LON-GNN`

- **OptBasisGNN**: `https://github.com/yuziGuo/FarOptBasis`

- **GNN-LF/HF**: `https://github.com/zhumeiqiBUPT/GNN-LF-HF`

- **ADA-UGNN**: `https://github.com/alge24/ADA-UGNN`

- **FE-GNN**: `https://github.com/sajqavril/Feature-Extension-Graph-Neural-Networks`

### E.3 HYPER-PARAMETERS SETTING

We perform a grid search on the hyper-parameters of all models (including baselines) for each dataset using the open-source package Optuna[3] (Akiba et al., 2019). To accommodate extensive experiments across diverse datasets in both semi- and full-supervised setting, we define a broad searching space as: learning rate lr $\sim$ {1e-3, 5e-3, 1e-2, 5e-2, 0.1}, weight decay $L_2 \sim$ {0.0, 1e-6, 5e-6, 1e-5, 5e-5, 1e-4, 5e-4, 1e-3, 5e-3, 1e-2}, dropout $\sim$ {0.0, 0.1, ..., 0.8} with step 0.1, non-local aggregation step $L \sim$ {1,2, ...,10} with step 1, scaling parameter $\tau \sim$ {0.1, 0.2, ..., 1.0} with step 0.1, update rate $\eta \sim$ {0.1, 0.2, ..., 1.0} with step 0.1, and threshold $\epsilon \sim$ {0.0, 1e-5, 5e-5, 1e-4, 5e-4, 1e-3, 5e-3, 1e-2}. For other parameters specific to different base models, we strictly follow their instructions in the original papers.

## F MORE EXPERIMENTS

### F.1 EMPIRICAL EVIDENCE OF THE ADAPTED NEW GRAPH

Additional empirical evidence related to the new graph's non-locality and signed edge weights can be found in Figure 5 and Figure 6, respectively.

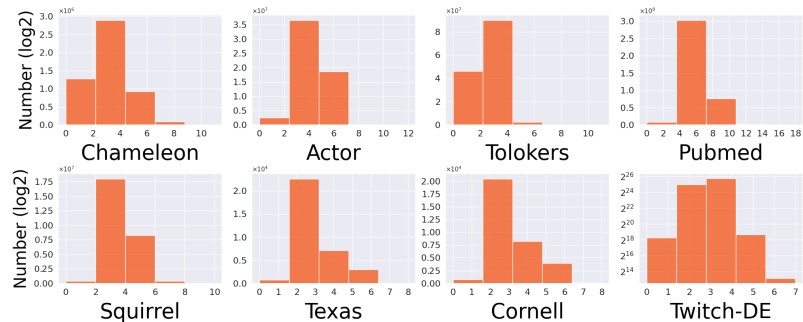

Figure 5: Additional distributions of original hop distance between adjacent nodes in the adapted new graph, where the number of bins is set as five for better histgram visualization.

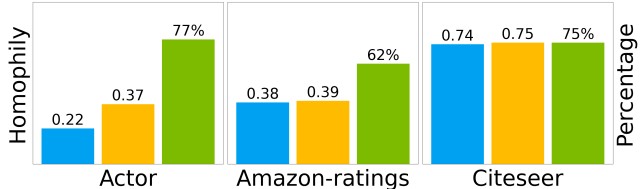

Figure 6: Additional visualizations for illustrating the newfound graph's ability in discerning label consistency. Blue, yellow, and green bars (sharing the same meanings as in the main text) respectively signifies the original graph, positive edges of new graph, and negative edges of new graph.

---

[3]`https://github.com/optuna/optuna`

## F.2 NEW BENCHMARKS FOR GRAPH HETEROPHILY.

For a more extensive evaluation across various domains, we also test SAF on the recently introduced datasets, including Minesweeper, Tolokers, Amazon-ratings, Roman-empire, and Twitch-DE. In this context, we draw comparisons solely with MLP, GCN, APPNP, along with six methods that have previously shown promising results in prior tasks, namely NodeFormer, GloGNN++, GPR-GNN, BernNet, ChebNetII, JacobiConv, and FE-GNN. Table 4 lists the average classification accuracies, obtained over 10 random splits as provided by (Platonov et al., 2023), with a distribution of 50%/25%/25% for training/validation/testing. To maintain consistency with this setting, we randomly divide the Twitch-DE dataset (Lim et al., 2021) ourselves. In summary, SAF achieves significant performance gains of 12.84% and 4.10% on Minesweeper and Tolokers, respectively. Meanwhile, SAF maintains competitiveness on the other three datasets, validating its reliability and effectiveness.

Table 4: Experimental results on new heterophilic graph datasets.

| Method | Minesweeper | Tolokers | Amazon-ratings | Roman-empire | Twitch-DE |
|---|---|---|---|---|---|
| MLP | 50.61±0.87 | 74.58±0.69 | 45.50±0.38 | 66.11±0.33 | 68.50±0.87 |
| GCN | 72.25±0.60 | 76.56±0.85 | 48.06±0.39 | 53.49±0.33 | 73.83±0.69 |
| APPNP | 68.48±1.20 | 74.13±0.62 | 48.12±0.37 | 72.99±0.46 | 72.81±0.66 |
| NodeFormer | 89.89±0.46 | **80.31±0.75** | 43.67±1.54 | 73.59±0.60 | 69.66±0.83 |
| GloGNN++ | 72.59±1.54 | 79.01±0.84 | 50.03±0.29 | 66.10±0.26 | 73.15±0.59 |
| GPR-GNN | 89.76±0.53 | 75.82±0.50 | 49.06±0.25 | 73.19±0.24 | 73.48±0.81 |
| ChebNetII | 83.62±1.51 | 78.95±0.49 | 49.76±0.36 | 74.52±0.54 | 74.17±0.68 |
| JacobiConv | 89.88±0.33 | 77.24±0.39 | 43.89±0.28 | 74.30±0.50 | 74.11±0.75 |
| FE-GNN | 84.68±0.36 | 79.31±0.59 | 49.46±0.29 | 74.50±0.30 | 73.60±0.60 |
| BernNet | 77.75±0.61 | 75.35±0.63 | 49.84±0.52 | 74.56±0.74 | 73.87±0.63 |
| SAF | 90.54±0.30 | 79.38±0.58 | **50.49±0.28** | 74.87±0.22 | **74.65±0.65** |
| SAF-$\epsilon$ | **90.59±0.35** | 79.45±0.60 | 50.36±0.15 | **74.89±0.29** | 74.55±0.68 |
| Improv. | 12.84% | 4.10% | 0.65% | 0.33% | 0.78% |

## F.3 ABLATION STUDY

Our SAF framework primarily comprises three modules: "Non-negative Spectral Filtering" (Spec.), "Non-local Spatial Aggregation" (Spat.), and "Node-wise Prediction Amalgamation" (Amal.). While the ablation study of the Spec. module is thoroughly analyzed in Section 6 of the main text, this appendix section extends this analysis by offering additional visualization results on more datasets, as depicted in Figure 7.

Focusing on the other modules, detailed results are presented in Table 5. These findings consistently show that excluding any module typically leads to a noticeable drop in performance, affirming the critical role each plays in our SAF framework. Notably, for the Amal. module, certain datasets, such as Cham. and Squi., demonstrate only a slight reduction in performance. This observation aligns with our observation that their optimal attention values are close to an even split, as suggested in Figures 3b and 8a.

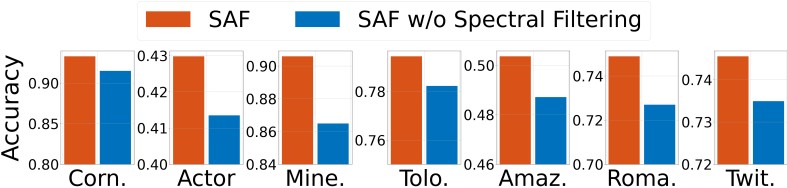

Figure 7: Ablation study of SAF w.r.t. Spectral Filtering on the additional seven datasets.

Table 5: Ablation study of SAF framework on 13 datasets regarding "Non-negative Spectral Filtering" (Spec.), "Non-local Spatial Aggregation" (Spat.) and "Node-wise Prediction Amalgamation" (Amal.) modules. For SAF w/o Amal., we remove the attention modules and blend predictions from different domains equally. As the SAF w/o Spat. configuration is equivalent to BernNet model, the corresponding results are posted directly.

| Our Variant | Cham. | Squi. | Texas | Corn. | Actor | Cora | Cite. | Pubm. | Mine. | Tolo. | Amaz. | Roma. | Twit. |
|---|---|---|---|---|---|---|---|---|---|---|---|---|---|
| SAF-$\epsilon$ | 74.84 | 64.00 | 94.75 | 93.28 | 42.98 | 89.87 | 81.45 | 91.52 | 90.59 | 79.45 | 50.36 | 74.89 | 74.55 |
| SAF | 75.30 | 63.63 | 94.10 | 92.95 | 42.93 | 89.80 | 80.61 | 91.49 | 90.54 | 79.38 | 50.49 | 74.87 | 74.65 |
| SAF w/o Amal. | 75.01 | 62.62 | 89.18 | 86.07 | 41.53 | 88.80 | 80.37 | 91.24 | 89.45 | 76.18 | 49.98 | 74.25 | 73.05 |
| SAF w/o Spec. | 73.55 | 55.70 | 90.49 | 88.20 | 41.06 | 88.03 | 78.87 | 90.12 | 89.45 | 78.23 | 49.13 | 71.85 | 73.54 |
| SAF w/o Spat. | 68.53 | 51.39 | 92.62 | 92.13 | 41.71 | 88.51 | 80.08 | 88.51 | 77.75 | 75.35 | 49.84 | 74.56 | 73.87 |

(a) Chameleon     (b) Minesweeper     (c) Tolokers     (d) Pubmed

Figure 8: Additional changing trends of the averaged attention values with respect to training epochs.

## F.4 ATTENTION TRENDS.

For the analysis of attention trends, we provide visualizations on more datasets in Figure 8. It is noteworthy that, despite Pubmed being a homophilic graph with $\mathcal{H} = 0.80$, non-local aggregation maintains a pivotal role in our SAF, diverging from the patterns observed on the Cora dataset as shown in Figure 3c. This discrepancy ownes much to the sizable node count within Pubmed, where the capability of non-local aggregation in capturing long-range dependencies proves advantageous for improving model performance, as indicated in Table 2.

## F.5 TIME AND SPACE OVERHEADS

**Spectral Decomposition.** Our SAF framework pre-computes spectral decomposition once per graph and reuses it for efficient matrix inversion in Eq. (6). This aspect is crucial, as the forward-pass cost in model training often exceeds the preprocessing expense of eigendecomposition. To empirically validate this, we compare the time overheads of spectral decomposition with the training times of various models in Table 6. It is evident that for most datasets, the time consumed by decomposition is significantly less than the time required for model training. For larger graphs such as Pubmed and Amazon-ratings, although the decomposition time is greater than that for local GNNs like BernNet and ChebNetII, it is still faster than the more advanced global GNNs including SAF, NodeFormer, and GloGNN++. In case of space overheads, similar patterns can be observed.

Table 6: Time overheads (s) / Space overheads (MB). The second and third column blocks represent local and global GNNs, respectively.

| Method | Cham. | Squi. | Texas | Corn. | Actor | Cora | Cite. | Pubm. | Amaz. |
|---|---|---|---|---|---|---|---|---|---|
| BernNet | 8.36/72 | 13.74/232 | 3.92/5 | 4.16/5 | 4.88/292 | 5.24/64 | 5.52/152 | 6.06/1546 | 13.98/2389 |
| ChebNetII | 22.82/72 | 30.73/231 | 11.47/5 | 9.64/5 | 14.88/291 | 19.96/63 | 16.14/152 | 36.91/1584 | 22.50/2355 |
| SAF | 11.55/112 | 18.78/440 | 4.38/5 | 4.70/5 | 5.36/733 | 6.04/120 | 6.12/237 | 18.43/4515 | 47.47/6966 |
| NodeFormer | 58.96/1522 | 79.66/3965 | 14.29/15 | 18.89/37 | 66.20/775 | 19.25/480 | 32.00/764 | 68.57/2119 | 122.91/3056 |
| GloGNN++ | 35.63/290 | 68.31/1525 | 4.47/5 | 3.00/5 | 73.13/2471 | 32.68/331 | 12.35/607 | 5266.53/17892 | 3614.37/25260 |
| Decomposition | 0.58/141 | 1.59/540 | 0.02/1 | 0.02/1 | 3.93 /1206 | 1.00/140 | 0.77 /239 | 21.34/7641 | 40.88/11442 |

**Model Comparison.** To evaluate the efficiency of our model, we compare the running times of our proposed variants, SAF and SAF-$\epsilon$, against four notable spectral GNNs (BernNet, JacobiConv,

Table 7: Average running time per epoch (ms)/average total running time (s)

| Method | Cham. | Squi. | Texas | Corn. | Actor | Cora | Cite. | Pubm. |
|---|---|---|---|---|---|---|---|---|
| BernNet | 14.60/8.36 | 17.40/13.74 | 16.10/3.92 | 14.30/4.16 | 14.40/4.88 | 14.90/5.24 | 16.20/5.52 | 15.10/6.06 |
| JacobiConv | 11.10/6.54 | 11.20/10.01 | 11.60/3.86 | 11.20/2.93 | 11.70/4.22 | 11.50/5.54 | 11.30/6.02 | 11.40/8.47 |
| ChebNetII | 39.30/22.82 | 40.70/30.73 | 42.30/11.47 | 39.60/9.64 | 41.80/14.88 | 39.20/19.96 | 40.70/16.14 | 40.60/36.91 |
| Specformer | 7.40/5.82 | 11.40/10.15 | 6.30/2.27 | 6.80/1.77 | 15.10/7.30 | 8.20/2.83 | 7.50/2.19 | 234.60/60.53 |
| NodeFormer | 135.00/58.96 | 135.10/79.66 | 49.10/14.29 | 67.60/18.89 | 158.20/66.20 | 56.70/19.25 | 73.10/32.00 | 150.60/68.57 |
| GloGNN++ | 53.60/35.63 | 123.70/68.31 | 14.80/4.47 | 9.80/3.00 | 204.80/73.13 | 89.60/32.68 | 49.60/12.35 | 6369.20/5266.53 |
| SAF | 19.30/11.55 | 20.90/18.78 | 17.70/4.38 | 18.60/4.70 | 18.80/5.36 | 18.90/6.04 | 18.10/6.12 | 43.30/18.43 |
| SAF-$\epsilon$ | 30.70/19.11 | 59.50/54.71 | 27.30/6.95 | 29.70/7.86 | 121.20/33.14 | 30.20/10.32 | 32.00/17.19 | 2054.80/837.10 |

ChebNetII, Specformer) and two cutting-edge non-local GNNs (NodeFormer, GloGNN++), as detailed in Table 7. One can see that SAF, while slightly slower than its base model, BernNet, due to the integration of non-local modules, remains more efficient than other leading non-local GNNs. Among other top spectral GNNs, SAF manages to achieve a comparable efficiency. On the other hand, SAF-$\epsilon$ costs much larger time, a consequence of its quadratic complexity from sparsification. However, this trade-off allows SAF-$\epsilon$ to produce an high-quality new graph, enhancing the efficiency of non-local aggregation and better capturing long-range interconnections and graph heterophily.

# G  ANALYSIS OF SAF WITH CHEBNETII AS BASE MODEL

To expand the versatility of our SAF framework, we introduced ChebNetII as an alternative base model, chosen for its adherence to the non-negative constraint critical in our model design. The rationale behind this choice is ChebNetII's use of Chebyshev interpolation to approximate Chebyshev polynomials, where ensuring non-negativity is simply a matter of keeping its learnable parameters $\{\gamma_j\}_{j=0}^K$ non-negative. Our experiments, as shown in Table 8, confirm that SAF can significantly enhances ChebNetII's performance, underscoring the framework's flexibility with different spectral filters.

Interestingly, we observed that SAF, when based on Bernstein polynomials (SAF-Bern), tends to slightly outperform SAF with ChebNetII (SAF-Cheb) in most datasets. The margin of improvement is also more pronounced with SAF-Bern. This phenomenon could be attributed to the $g_\phi(\lambda) \leq 1$ constraint within SAF (refer to Section 5), necessitating the rescaling of filter functions by their maximum graph spectrum values. For Bernstein polynomials, this maximum is readily obtained as the largest polynomial coefficient ($\max\{\phi_k\}_{k=0}^K$, as per Proposition 3). However, for Chebyshev polynomials, the best theoretical upper bound is the sum of absolute coefficients, $\sum_{k=0}^K |\phi_k|$, which is comparatively less precise. This difference may impact the quality of graph construction and, subsequently, the model's performance. Exploring these nuances will be a focal point of our future research.

Table 8: Full-supervised node classification accuracies (%). SAF-Bern and SAF-Cheb refers to SAF implementation respectively using BernNet and ChebNetII as base model.

| Method | Cham. | Squi. | Texas | Corn. | Actor | Cora | Cite. | Pubm. |
|---|---|---|---|---|---|---|---|---|
| BernNet | 68.53±1.68 | 51.39±0.92 | 92.62±1.37 | 92.13±1.64 | 41.71±1.12 | 88.51±0.92 | 80.08±0.75 | 88.51±0.39 |
| SAF-Bern | 75.30±0.96 | 63.63±0.81 | 94.10±1.48 | 92.95±1.97 | 42.93±0.79 | 89.80±0.69 | 80.61±0.81 | 91.49±0.29 |
| Improv. | 6.77% | 12.24% | 1.48% | 0.82% | 1.22% | 1.29% | 0.53% | 2.98% |
| ChebNetII | 71.37±1.01 | 57.72±0.59 | 93.28±1.47 | 92.30±1.48 | 41.75±1.07 | 88.71±0.93 | 80.53±0.79 | 88.93±0.29 |
| SAF-Cheb | 74.97±0.66 | 64.06±0.59 | 94.43±1.81 | 92.62±2.13 | 42.65±1.01 | 89.56±0.64 | 80.68±0.68 | 91.27±0.34 |
| Improv. | 3.60% | 6.34% | 1.15% | 0.32% | 0.90% | 0.85% | 0.15% | 2.34% |

# H  PARAMETER ANALYSIS

In this section, we delve into the sensitivity of four important hyper-parameters: $\tau$, $\eta$, $\epsilon$, and $L$. The learning performance variations with different parameter selections from a wide range are illustrated in Figure 9. Notably, our SAF model demonstrates robust stability across an extensive range of parameter values. For example, choosing values for $\tau$ and $\eta$ from the set $\{0.1, 0.2, ..., 1\}$ yields promising results. These ranges are in line with our parameter tuning strategy as detailed in Appendix E.3.

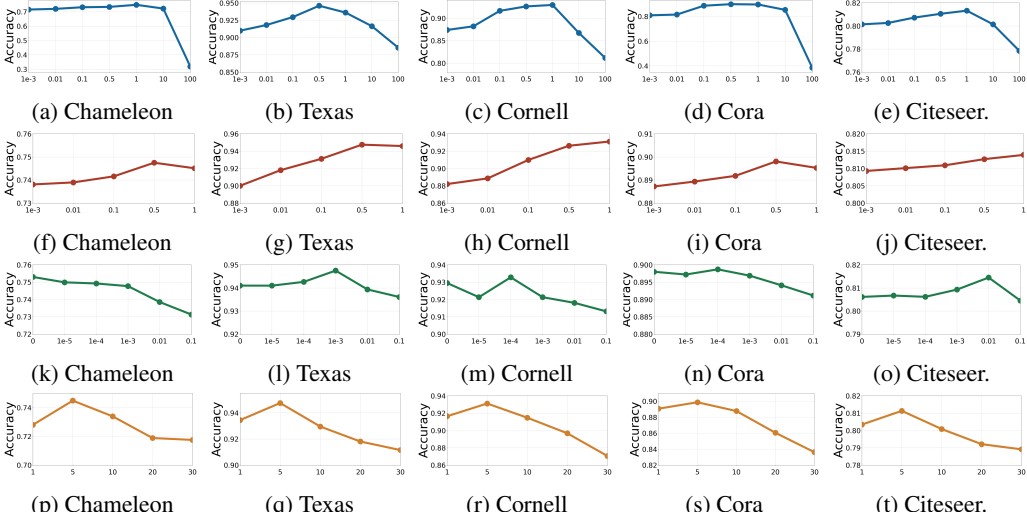

Figure 9: Sensitivity analysis of hyper-parameters (top to bottom): $\tau$, $\eta$, $\epsilon$, and $L$, each respectively annotated in blue, red, green, and orange.

Particularly for the non-local aggregation layer number $L$, a noticeable decline in model performance is observed when $L$ exceeds 10. This is attributed to the non-local nature of our new graph, which facilitates efficient information exchange between nodes. Exceeding a certain number of layers may potentially lead to oversmoothing, where there is an overemphasis on global information, thus degrading model performance. However, choosing the number of layers within a reasonable range generally ensures consistent and impressive model performance, as verified in Figures 9p- 9t.

