# OpenReview forum: "Rethinking Spectral Graph Neural Networks with Spatially Adaptive Filtering"
_ICLR.cc/2024/Conference — Submitted to ICLR 2024_

### Official Review · Reviewer_k2v7 · 2023-10-19

**Soundness:** 3 good
**Presentation:** 3 good
**Contribution:** 3 good
**Rating:** 5
**Confidence:** 3

**Summary:**

This paper studies the connection between spectral and spatial domains for graph neural networks (GNN). Starting from a graph optimization problem, the authors can interpret an adapted new graph (used for propagation) based on the graph filter. Due to the inversion operation applied to the graph filter and the range of the graph filter’s eigenvalues, this new graph can express an infinite series of the graph Laplacian. Motivated by this advantage against truncated polynomial approximators, the authors design a novel paradigm of GNN, named SAF, which utilizes BernNet and attends to the local and global information. Extensive experiments show the advantages of SAF on various datasets and under different settings.

**Strengths:**

1.	This paper is well-written. I can pick the core idea effortlessly.
2.	The proposed paradigm SFA is theoretically motivated, and its advantages seem to be well attributed to the expressivity of the adapted new graph.
3.	The empirical studies are comprehensive. It makes the results convincing to evaluate related methods on those recently introduced node classification datasets. The results in the ablation study confirm the source of end2end advantages is the more expressive spectral filtering.

**Weaknesses:**

1.	In my opinion, the most salient limitation of this work is the necessity to compute spectral decomposition. Although there have been many acceleration methods to conquer the O($N^3$) complexity (as the authors have also mentioned), such complexity will make SFA inapplicable to even moderate-sized graphs such as ogbn-products. Besides, there have been many more theoretical yet less practical graph filtering methods, where what limits their applications is mainly the necessity of spectral decomposition.
2.	I am a little confused with the so-called supervised setting. It is obvious that the split ratio is different from that of a semi-supervised setting. However, it seems that the node features of test nodes are also accessible to the compared methods during the training stage, which is the same as the semi-supervised setting.
3.	Theoretically, as the adapted new graph can express the power series of graph Laplacian, the most straightforward advantage of SFA, against existing spectral GNNs that use truncated polynomial approximation, is the capability of fitting specific graph filters. Thus, it is desirable to include such experiments in the main context of this paper as BernNet.

**Questions:**

Could you explain figure 1 for me? I cannot figure out what the x-axis means as well as how these figures imply your conclusion.

---

> ### Author Response · Authors · 2023-11-17
>
> > Q.1  Limitations regarding spectral decomposition in this work.
>
> Thank you for your feedback on our use of spectral decomposition. We have addressed this shared concern in our "global" response.
>
>
> > Q.2 I am a little confused with the so-called supervised setting.
>
> Thank you for your inquiry about the "supervised setting" in our experiments.
>
> **(1)** In line with established experimental protocols in transductive node classification, as referenced in prior works [1,2,3], our approach utilizes all node features (train, validation, and test) while only incorporating training node labels during the training phase. This method is a standard practice in the field, aimed at evaluating GNNs in node-level tasks.
>
> **(2)** To further evaluate models comprehensively under varying levels of supervision, we implemented two distinct data splitting strategies: a sparse splitting (2.5%/2.5%/95%) and a dense splitting (60%/20%/20%) for the training, validation, and testing sets, respectively. Consistent with the terminology used in [3], we refer to these tasks as "semi-supervised" and "full-supervised" node classification. This distinction is based on the degree of supervision each setting entails.
>
> [1] Chien, Eli, et al. "Adaptive universal generalized pagerank graph neural network." ICLR, 2021.
>
> [2] He, Mingguo, et al. "Bernnet: Learning arbitrary graph spectral filters via bernstein approximation." NeurIPS, 2021.
>
> [3] He, Mingguo, et al. "Convolutional neural networks on graphs with chebyshev approximation, revisited." NeurIPS, 2022.
>
>
>
> > Q.3  Theoretically, as the adapted new graph can express the power series of graph Laplacian, the most straightforward advantage of SAF, against existing spectral GNNs that use truncated polynomial approximation, is the capability of fitting specific graph filters.
>
>
> Thanks for your constructive feedback and the insightful suggestion to incorporate experiments on fitting specific filters. We appreciate this opportunity to clarify the scope and focus of our work.
>
> **(1) Essence of the Adapted New Graph:** In our theoretical framework, the adapted new graph is fundamentally anchored in the spatial domain. This spatial orientation plays a pivotal role in making spectral GNNs interpretable. While it is true that our new graph expresses the infinite power series of the graph Laplacian (more so of the original adjacency matrix $\hat{\mathbf{A}}$), this aspect mainly highlights its non-local nature in the vertex domain. This non-locality is crucial for our approach to capture complex node relations, but it does not intrinsically boost expressiveness in the spectral domain.
>
> **(2) Enhancement of Spectral GNNs by SAF:** Our SAF framework is designed to address a key limitation of existing spectral GNNs -- the oversight of global vertex dependencies caused by truncated polynomials. By employing spatial aggregation on the non-local new graph, SAF adeptly captures long-range dependencies between nodes, a feature we have rigorously demonstrated in our experiments. These findings demonstrate SAF's effectiveness in handling complex graph structures, particularly in scenarios like heterophilic graphs where global interactions are essential.
>
> **(3) Comparison in Fitting Specific Filters:** Addressing the spectral domain challenge of fitting specific filters, it's noteworthy that both our SAF and BernNet utilize Bernstein polynomials as their spectral filter functions. This commonality implies that any experimental results in this regard would likely be identical for both frameworks. Hence, we chose not to include this specific task in our experiments. The congruence in the outcomes also underscores that our framework aligns with the capabilities of existing approaches in the spectral domain.
>
>
> > Q.4 Could you explain figure 1 for me?
>
> Thanks for your question regarding Figure 1 in our paper. We are pleased to provide further clarification on its content and how it supports our conclusions.
>
> **(1)** In Figure 1, the x-axis represents the geodesic distance between node pairs in the original graph. This distance is calculated as the number of hops in the shortest path between the vertices. Concurrently, the y-axis quantifies the frequency of node pairs (of various types) being connected in the new graph.
>
> **(2)** Using the Minesweeper dataset in Figure 1, where the longest geodesic distance is 99, as an example, we observe that node pairs from the original graph, regardless of their distance, are often directly connected in the new graph (indicated by a frequency greater than 0). This finding is crucial as it demonstrates the capability of the new graph to establish direct links among nodes that originally require multiple hops to do so.
>
> **(3)** To enhance clarity, we have revised the caption of Figure 1 as "Distributions of adjacent nodes in the new graph based on their geodesic distance in the original graph. It is seen that nodes, distant in the original graph, can be connected in the new graph.".

---

> ### Comment · Reviewer_k2v7 · 2023-11-21
> **Discussion**
>
> Thanks for your detailed explanation! I also read your general response. I appreciate the analysis and the connections established in this paper. However, when it comes to instantiate a concrete GNN architecture, it requires the eigen-decomposition, which is unaffordable for even graph of moderate size. This is not a severe issue for a research paper in my opinion, but there would be a lot of interesting design/variants for spectral GNNs if such a computation is allowed. In this sense, I tend to keep the rating unchanged, while raising contribution to 3.

---

> > ### Author Response · Authors · 2023-11-22
> > **Further Clarification**
> >
> > We thank you for going through our rebuttal and acknowledging the strengths of our work by raising the contribution score. We are happy that our responses have addressed your previous concerns. While we are encouraged by your positive feedback, we would like to seek some clarification regarding your decision to keep the current overall rating, particularly in light of your recognization that our usage of spectral decomposition is not a significant issue for a research paper.
> >
> > In response to the perspective you kindly shared:
> >
> > > This is not a severe issue for a research paper in my opinion, but there would be a lot of interesting design/variants for spectral GNNs if such a computation is allowed.
> >
> > We would like to offer some additional insights:
> >
> > **(1)** We fully agree with your opinion that allowing spectral decomposition could open avenues for interesting designs and variants in spectral GNNs. Indeed, our paper already references over a dozen  works published in ICLR, NeurIPS, and ICML that have advanced GNNs, both theoretically and practically, by employing explicit eigendecomposition (see Appendix C.3).
> >
> > **(2)** In particular, we discuss two of these works (FE-GNN and Specformer) using spectral decomposition, showcasing their relevance to our study, respectively in the “Unified Viewpoints for GNNs” and “Long-range Dependencies” subsections of Section 2. We also include these two models in our empirical evaluations (see Tables 1 and 2).
> >
> > **(3)** We mainly compare our SAF with SoTA polynomial spectral GNNs that avoid decomposition operations (such as GPR-GNN, BernNet, ChebNetII, JacobiConv, LON-GNN, and OptBasisGNN). Our empirical findings demonstrate that SAF outperforms these models across multiple datasets, combined with enhanced interpretability, suggesting the computational requirement of spectral decomposition are well-compensated.
> >
> > **(4)** As for the computational aspects, spectral decomposition in our approach is a one-time computation per graph, which can be efficiently reused. Its preprocessing cost is often lower than the forward-pass cost in model training, as confirmed by our experiments in the global response. For larger graphs, while the decomposition process may be slower than training some local GNN models, it remains competitively efficient compared to global GNN models, both in terms of time and space overheads.
> >
> > We sincerely hope that these points address your concerns. Your insights are invaluable to us, and we remain open and eager for any further discussion or feedback you might have.

---

### Official Review · Reviewer_wm9b · 2023-10-30

**Soundness:** 3 good
**Presentation:** 3 good
**Contribution:** 3 good
**Rating:** 6
**Confidence:** 4

**Summary:**

The paper examines spectral GNNs from a spatial perspective, uncovering their interpretability in the spatial domain. It reveals that spectral GNNs implicitly transform the original graph into an adapted version, incorporating non-locality and signed edge weights for label consistency. Based on these insights, the authors propose a Spatially Adaptive Filtering framework that enhances spectral GNNs by leveraging the adapted graph structure for non-local aggregation. The SAF framework addresses issues related to long-range dependencies and graph heterophily by considering global node similarity and dissimilarity. Extensive experiments on 13 node classification benchmarks demonstrate the superiority of the proposed SAF framework over existing models. This work provides valuable understanding of spectral GNNs in the spatial domain and offers a promising approach for improving graph representation learning.

**Strengths:**

1.	Connect spatial and spectral GNNs. Spectral filters modify the original graph into a new graph showing nice non-locality.
2.	It proposes a novel Spatially Adaptive Filtering (SAF) framework. SAF can balance between spectral and spatial features. As a result, it can mitigate two problems of GNNs: long-range dependencies and graph heterophily.
3.	It provides an analysis about the newly adapted graph. There are two properties: Non-locality and Discerning Label Consistency, which alleviates the long-range dependency and heterophily problems in graphs.
4.	Comprehensive experiments show that the proposed SAF can promote the performance of BernNet on various datasets.

**Weaknesses:**

1.	SAF is only compatible with BernNet due to its assumption of non-negative filtering. As a result, it raises questions about whether the observed good performance is attributable to BernNet or SAF?

2.	When compared to other spectral methods, SAF requires eigen decomposition, which incurs a time complexity of O(n^3). This is a computational cost that exceeds that of other baseline methods that does not need eigen decomposition (BernNet, ChebNetII).

**Questions:**

See Weaknesses.

---

> ### Author Response · Authors · 2023-11-17
>
> > Q.1  SAF is only compatible with BernNet due to its assumption of non-negative filtering. As a result, it raises questions about whether the observed good performance is attributable to BernNet or SAF?
>
>
> Thank you for your thoughtful feedback regarding the compatibility of our SAF framework with BernNet. We appreciate the opportunity to clarify and expand on this important aspect of our work.
>
> **(1)** As you rightly pointed out, SAF indeed requires non-negative graph filters, which we discuss in Section 5 of our paper. However, we would like to highlight that SAF's compatibility extends beyond BernNet. **Theoretically, SAF is also applicable to other models like ChebNetII [1], a recent Spectral GNN variant based on Chebyshev interpolation, by reparameterizing its learnable parameters to be non-negative.** To substantiate this theoretical proposition, we have conducted additional experiments with ChebNetII, incorporating our SAF framework. These experiments yielded encouraging results as shown in the following table, proving the potential of SAF in enhancing models other than BernNet.
>
> Full-supervised node classification accuracies (\%)
> |              |   Cham.            |   Squi.            |   Texas            |   Corn.            |   Actor            |   Cora             |   Cite.            |   Pubm.            |
> |--------------|--------------------|--------------------|--------------------|--------------------|--------------------|--------------------|--------------------|--------------------|
> |   ChebNetII  |   71.37$\pm$1.01   |   57.72$\pm$0.59   |   93.28$\pm$1.47   |   92.30$\pm$1.48   |   41.75$\pm$1.07   |   88.71$\pm$0.93   |   80.53$\pm$0.79   |   88.93$\pm$0.29   |
> |   SAF-Cheb   |   74.97$\pm$0.66   |   64.06$\pm$0.59   |   94.43$\pm$1.81   |   92.62$\pm$2.13   |   42.65$\pm$1.01   |   89.56$\pm$0.64   |   80.68$\pm$0.68   |   91.27$\pm$0.34   |
> |   Improv.    |   3.60             |   6.34             |   1.15             |   0.32             |   0.90             |   0.85             |   0.15             |   2.34             |
>
> **(2)** While SAF, as detailed in our paper, does build upon the foundational structure of BernNet, **its distinct feature is the inclusion of non-local aggregation, aimed at capturing long-range dependencies and addressing graph heterophily.** This aspect is crucial for advanced graph learning and is not a native feature of BernNet. Our experimental results, as detailed in Tables 1, 2, and 4, clearly show that the enhancements brought by SAF are distinct and significant, with performance improvements of up to 15.37\%. These results underscore the unique contributions of SAF, reaffirming its significance beyond the inherent capabilities of BernNet.
>
> [1] He, Mingguo, et al. "Convolutional neural networks on graphs with chebyshev approximation, revisited." NeurIPS, 2022.
>
> > Q.2  When compared to other spectral methods, SAF requires eigen decomposition, which incurs a time complexity of O(n 3). This is a computational cost that exceeds that of other baseline methods that does not need eigen decomposition (BernNet, ChebNetII).
>
> Thanks for raising the question regarding our explict use of spectral decomposition. As this concern was shared among multiple reviewers. we've addressed it in the "global" response for a thorough explanation.

---

> > ### Comment · Reviewer_wm9b · 2023-11-23
> >
> > Thanks for your rebuttal. Most of my concerns have been solved. I will keep my overall score.

---

### Official Review · Reviewer_WZnB · 2023-10-30

**Soundness:** 3 good
**Presentation:** 3 good
**Contribution:** 3 good
**Rating:** 6
**Confidence:** 5

**Summary:**

This paper explores the significance of spectral graph neural networks (GNNs) in the spatial domain, highlighting their role in capturing non-local information and label consistency. To address limitations associated with fixed-order polynomials, the authors introduce the Spatially Adaptive Filtering (SAF) framework, enabling flexible integration of spectral and spatial features. SAF overcomes issues related to truncated polynomials, enhancing the model's ability to handle long-range dependencies and graph heterophility. Experimental results demonstrate substantial improvements, with SAF outperforming other spectral GNNs on average across various benchmarks.

**Strengths:**

1) In this paper, the concept of bridging spectral GNNs and spatial GNNs via a graph optimization framework is a novel and innovative idea.
2) The SAF framework introduced in this paper enhances the performance of BernNet, surpassing the performance of other baseline models in scenarios involving heterophilic graphs.
3) Interestingly, the presence of negative edges in the new graph can provide insights into label consistency within heterophilic graphs.
4) The experiments conducted in this paper are impressively comprehensive, encompassing a wide array of datasets, baseline comparisons, ablation studies, and visualizations.

**Weaknesses:**

1. My main concern is the use of explicit eigndecomposition, which I believe is unacceptable in spectral GNNs.
- The time and space complexity of eigendecomposition for an $n$-dimensional square matrix is $O(n^3)$ and $O(n^2)$, respectively, severely limiting scalability in practice. Even as a preprocessing step, it becomes challenging to execute on graphs with millions of nodes and edges.
- The primary advantage of spectral GNNs based on polynomials is their ability to avoid the need for eigendecomposition. If eigenvectors were readily available, the design of spectral GNNs would become trivial, rendering polynomial-based approaches unnecessary.
- While certain works, such as Specformer, have used eigendecomposition, I believe their practical significance is quite limited.
- Although the authors have conducted practical complexity experiments, preprocessing time and space complexity have not been included.

2. Opting for the Bernstein basis as the backbone may not represent the optimal choice. On the one hand, ChebNetII can also guarantee filter non-negativity, and on the other hand, Chebyshev interpolation boasts lower complexity in comparison to Bernstein approximation.

**Questions:**

1) Please refer to the aforementioned weaknesses.
2) My main concern is the practicality of explicit eigendecomposition, and I would welcome a discussion between the author and other reviewers on this issue.

---

> ### Author Response · Authors · 2023-11-17
>
> > Q.1  Computational Concerns about the use of explicit eigndecomposition.
>
> Thank you for raising this important question. We acknowledge that this query was shared among multiple reviewers. To provide a comprehensive response, we have elaborated on this topic in our "global" response section.
>
>
>
> > Q.2  Opting for the Bernstein basis as the backbone may not represent the optimal choice. On the one hand, ChebNetII can also guarantee filter non-negativity, and its Chebyshev interpolation boasts lower complexity in comparison to Bernstein approximation.
>
> Thank you for your insightful comments regarding our choice of the Bernstein basis in our SAF framework. Your comment has prompted a valuable extension of our analysis, which can be found in the Appendix G of our revised manuscript.
>
> **(1) Experimenting with ChebNetII as Base Model:** We sincerely apologize for the initial omission in not including experiments with ChebNetII as a base model for SAF. We have now conducted additional experiments with ChebNetII as a base model for SAF (SAF-Cheb), alongside our original implementation using BernNet (SAF-Bern). The results, which we present in the subsequent table, illustrate that SAF indeed enhances ChebNetII’s performance.
>
> Full-supervised node classification accuracies (\%)
> |              |   Cham.            |   Squi.            |   Texas            |   Corn.            |   Actor            |   Cora             |   Cite.            |   Pubm.            |
> |--------------|--------------------|--------------------|--------------------|--------------------|--------------------|--------------------|--------------------|--------------------|
> |   BernNet    |   68.53$\pm$1.68   |   51.39$\pm$0.92   |   92.62$\pm$1.37   |   92.13$\pm$1.64   |   41.71$\pm$1.12   |   88.51$\pm$0.92   |   80.08$\pm$0.75   |   88.51$\pm$0.39   |
> |   SAF-Bern   |   75.30$\pm$0.96   |   63.63$\pm$0.81   |   94.10$\pm$1.48   |   92.95$\pm$1.97   |   42.93$\pm$0.79   |   89.80$\pm$0.69   |   80.61$\pm$0.81   |   91.49$\pm$0.29   |
> |   Improv.    |   6.77             |   12.24            |   1.48             |   0.82             |   1.22             |   1.29             |   0.53             |   2.98             |
> |   ChebNetII  |   71.37$\pm$1.01   |   57.72$\pm$0.59   |   93.28$\pm$1.47   |   92.30$\pm$1.48   |   41.75$\pm$1.07   |   88.71$\pm$0.93   |   80.53$\pm$0.79   |   88.93$\pm$0.29   |
> |   SAF-Cheb   |   74.97$\pm$0.66   |   64.06$\pm$0.59   |   94.43$\pm$1.81   |   92.62$\pm$2.13   |   42.65$\pm$1.01   |   89.56$\pm$0.64   |   80.68$\pm$0.68   |   91.27$\pm$0.34   |
> |   Improv.    |   3.60             |   6.34             |   1.15             |   0.32             |   0.90             |   0.85             |   0.15             |   2.34             |
>
> **(2) Performance Comparison:** Interestingly, SAF-Bern slightly outperforms SAF-Cheb in most datasets, and the improvement margin with SAF-Bern is notably larger. This may be due to the $g_ \phi(\lambda) \leq 1$ constraint in our SAF (see Section 5), where filter functions need to be rescaled by their maximum values on the graph spectrum. For Bernstein polynomials, this maximum is simply the maximum polynomial coefficient ($\max \{\phi_k\}_ {k=0}^K$), as per our Proposition 3. In contrast, the best theoretical upper bound for Chebyshev polynomials is $\sum_ {k=0}^K |\phi_ k|$, which is less precise. This distinction potentially affects the new graph's construction, thereby limiting model performance. We plan to address this issue in future work.
>
> **(3) Empirical Complexity Studies:** We acknowledge that Chebyshev interpolation boasts greater efficiency over Bernstein approximation in theory. However, our empirical studies indicate that BernNet and SAF-Bern generally demand less training time compared to ChebNetII and SAF-Cheb, as evidenced in the table below. This could be due to ChebNetII requiring additional computations for the values w.r.t. $(K+1)^2$ Chebyshev nodes.
>
> Average total running time (s).
> |               |   Cham.  |   Squi.  |   Texas  |   Corn.  |   Actor  |   Cora   |   Cite.  |   Pubm.  |
> |---------------|----------|----------|----------|----------|----------|----------|----------|----------|
> |   BernNet     |   8.36   |   13.74  |   3.92   |   4.16   |   4.88   |   5.24   |   5.52   |   6.06   |
> |   ChebNetII   |   22.82  |   30.73  |   11.47  |   9.64   |   14.88  |   19.96  |   16.14  |   36.91  |
> |   SAF-Bern    |   11.55  |   18.78  |   4.38   |   4.70   |   5.36   |   6.04   |   6.12   |   18.43  |
> |   SAF-Cheb    |   19.29  |   19.87  |   8.31   |   7.97   |   13.27  |   10.65  |   19.22  |   39.94  |

---

> > ### Comment · Reviewer_WZnB · 2023-11-20
> > **Response**
> >
> > Thank you for the author's responses and the extra experimental results. I believe the method proposed in this paper is effective and sound. My main concern is whether it is reasonable to promote the use of eigendecomposition in current graph convolutional networks, considering the challenges of applying these methods to graphs with over a million nodes. However, given that many previous works have already employed eigendecomposition and taking into account the contributions made by this paper, I am inclined to increase my rating to 6 and lean towards accepting this paper.

---

> > > ### Author Response · Authors · 2023-11-22
> > >
> > > We sincerely thank you for your thoughtful review and for increasing the score of our paper. We are glad to know that our responses, particularly regarding spectral decomposition, have addressed your concerns and supported your inclination towards acceptance. Please let us know if there are further questions or unresolved issues. We are very willing to clarify.

---

### Official Review · Reviewer_2DDa · 2023-11-01

**Soundness:** 4 excellent
**Presentation:** 3 good
**Contribution:** 3 good
**Rating:** 5
**Confidence:** 3

**Summary:**

This paper investigated the relationship between spectral graph filters and spatial GNNs. The theoretical results suggest that the graph filter obtained in spectral space is able to construct a non-local new graph that contains global connectivity information. By combining the representation aggregating from this adapted new graph with the spectral embedding obtained from polynomial graph filters, a new framework SAF is proposed for node classification.
The experiments revealed what the adapt adjacency learned in spatial space. The ablation study shows the effectiveness of spectral filtering, and the node classification performance of SAF is competitive.

**Strengths:**

1. Overall, this paper is theoretically solid. The relationship between spatial and spectral is clearly uncovered with theoretical proofs.
2. The experimental analysis of SAF can effectively demonstrate its capability for capturing heterophilous edges and punishing them with negative weights.
3. The motivation for finding the correlation between spatial and spectral and combining them is clear and intriguing.
4. The analysis of attention trends is consistent with intuition, showing the effectiveness of the proposed SAF.

**Weaknesses:**

Although the paper includes impressive theoretical analyses, there are some weaknesses as below:

1. The correlation between spectral and spatial domain graph learning has been exposed by several previous works although they do not emphasize this. For example, the work cited in your paper, Interpreting and unifying graph neural networks with an optimization framework, has derived both spectral-domain graph filters and spatial-domain GNN frameworks from the denoising optimization target; some other works like GCNII have also endeavored to analyze their spatial GNN model in the spectral domain.

2. The authors claim that the polynomial filters are limited in K hop, but they do not discuss how this problem is addressed by the proposed model. In my view, it is hard to understand why the combination of graph filter and adapted adjacent also derived from this graph filter can address this limitation.

3. The ablation study should exclude each component.

4. This paper lacks of parameter sensitive analysis regarding $\tau, \eta, \epsilon$, and the experimental analysis regarding different $L$ and $K$.

**Questions:**

Please try to fix the questions stated above.

---

> ### Author Response · Authors · 2023-11-17
> **Response (1/3)**
>
> > Q.1  Previous works on the correlation between spectral and spatial domain graph learning.
>
> Thanks for your insightful comments. We are glad to further clarify our work's position in relation to previous studies on the correlation between spectral and spatial domain graph learning.
>
> **(1)** In our original submission, we actually have acknowledged these pioneering researches, including but not limited to works [1,2,3], in exploring the congruencies between spectral and spatial GNNs. This discussion is present in the second paragraph of Section 1 and the "Unified Viewpoints for GNNs" subsection of Section 2. **Our work, distinct from these previous efforts, represents the first endeavor to delve into the interpretability of spectral GNNs from the spatial domain, examining the theoretical interplay between spectral filtering and spatial aggregation.**
>
> **(2)** Regarding GCNII [4], which utilizes spectral analysis to validate its anti-oversmoothing capability, we regret its omission in our earlier discussion and have now included it in Section 2 (highlighted in blue).
>
> **(3)** In our experiments, we have compared our SAF framework against relevant models including GNN-LF [1], GNN-HF [1], ADA-UGNN [2], FE-GNN [3], and GCNII [4], as detailed in Tables 1 and 2 of our manuscript. **These comparisons demonstrate that SAF significantly outperforms these models (including the two as you kindly mentioned)**, thereby underscoring our contribution to the existing literature in not only theoretical but also practical terms.
>
> [1] Zhu, Meiqi, et al. "Interpreting and unifying graph neural networks with an optimization framework." WWW, 2021.
>
> [2] Ma, Yao, et al. "A unified view on graph neural networks as graph signal denoising." CIKM, 2021.
>
> [3] Sun, Jiaqi, et al. "Feature expansion for graph neural networks." ICML, 2023.
>
> [4] Chen, Ming, et al. "Simple and deep graph convolutional networks." ICML, 2020.
>
> > Q.2 Why the combination of polynomail graph filter and the adapted adjacent also derived from this filter can address its limitation?
>
> Thanks for your insightful comments. We appreciate the opportunity to clarify the capabilities of our SAF framework in addressing the limitations of spectral GNNs.
>
> **(1)** Most spectral GNNs are limited by their reliance on fixed-order polynomials for graph filter approximation, confining their effective propagation to within $K$ hops and thus restricting long-range dependency capture. Our research, however, has unveiled an intriguing aspect. We discovered that spectral filtering, while performed in the spectral domain, implicitly crafts a non-local new graph in the spatial domain. This new graph, free from the constraints of the original topology, uncovers deeper, previously unobserved node connections. **Despite this, domain barriers impede spectral GNNs from exploiting this non-locality existing in the spatial domain.** In response, our SAF framework is designed to enhance spectral filtering with auxiliary non-local aggregation. This combination effectively overcomes the truncated polynomials' limitations, enriching long-range dependency capture and graph information exploration.
>
> **(2)** This concept is extensively discussed in the last paragraph of Section 4.1 of our paper, where we elaborate on the distinct mechanics of the adapted new graph compared to conventional spectral filtering. Moreover, in Section 6, particularly in the "Full-supervised Node Classification" subsection, we provide empirical evidence and analytical insights that demonstrate SAF's superiority over other spectral GNNs, substantiating our methodology.

---

> ### Author Response · Authors · 2023-11-17
> **Response (2/3)**
>
> > Q.3 The ablation study should exclude each component.
>
> Thank you for your valuable suggestion to augment our ablation analysis. Your guidance has been instrumental in enhancing the comprehensiveness of our study, and we have added detailed analysis in the revision (see Appendix F.3).
>
> **(1)** Our SAF framework primarily comprises three modules: "Non-negative Spectral Filtering" (Spec.), "Non-local Spatial Aggregation" (Spat.), and "Node-wise Prediction Amalgamation" (Amal.). We have already tested our SAF without the Spec. module in the "Ablation Study" subsection of Section 6. Additionally, the configuration of SAF w/o Spat. is effectively equivalent to the BernNet model, of which the performance can be observed in Tables 1 and 2.
>
> **(2)** Following your kind suggestion, we have now included experiments that ablate the Amal. module. We implemented this by removing the attention mechanism and equally blending predictions from different domains. To provide a comprehensive view, all ablation results are presented in the following table. These results illustrate that the omission of any module generally results in a noticeable decrease in performance, affirming the effectiveness of our design. Interestingly, for the Amal. module, some datasets like Cham. and Squi. exhibit only a modest performance decline. This observation aligns with our observation that their optimal attention values are close to an even split, as suggested in Figures 3(b) and 8(a).
>
> Ablation study of SAF framework regarding “Node-wise Prediction Amalgamation” (Amal.), “Non-local Spatial Aggregation” (Spat.) and “Non-negative Spectral Filtering” (Spec.) modules.
> |                  |   Cham.  |   Squi.  |   Texas  |   Corn.  |   Actor  |   Cora   |   Cite.  |   Pubm.  |
> |------------------|----------|----------|----------|----------|----------|----------|----------|----------|
> |   SAF-$\epsilon$  |   74.84  |   64.00  |   94.75  |   93.28  |   42.98  |   89.87  |   81.45  |   91.52  |
> |   SAF            |   75.30  |   63.63  |   94.10  |   92.95  |   42.93  |   89.80  |   80.61  |   91.49  |
> |   SAF w/o Amal.  |   75.01  |   62.62  |   89.18  |   86.07  |   41.53  |   88.80  |   80.37  |   91.24  |
> |   SAF w/o Spec.  |   73.55  |   55.70  |   90.49  |   88.20  |   41.06  |   88.03  |   78.87  |   90.12  |
> |   SAF w/o Spat.  |   68.53  |   51.39  |   92.62  |   92.13  |   41.71  |   88.51  |   80.08  |   88.51  |

---

> ### Author Response · Authors · 2023-11-17
> **Response (3/3)**
>
> > Q.4 This paper lacks of parameter analysis regarding $\tau, \eta, \epsilon$, $L$ and $K$.
>
> Thanks for your insightful feedback regarding the sensitivity analysis of hyper-parameters in our model. We regret the initial omission of this crucial analysis in our experiments. To address this, we have now conducted a comprehensive analysis of our model's behavior across a wide range of hyper-parameter choices, specifically for $\tau,
> \eta, \epsilon$, and $L$. More detailed results and analysis have been added in the revision (see Appendix H).
>
> **(1)** Regarding the parameter $K$, which represents the number of polynomial orders in the spectral filter, **we aligned with existing literature and fixed it at 10 in our SAF model.** This choice reflects our focus on the novel aspects of SAF rather than on parameters well-explored in previous works.
>
> **(2)** To provide a clear illustration of our findings, we have included results from two datasets, Texas and Cora, in the following tables as an typical example. **As can be observed, our SAF model exhibits robust stability across a broad range of parameter values.** For instance, promising performance can be obtained by selecting both $\tau$ and $\eta$ values from {$0.1, 0.2, ..., 1$}. This interval aligns with the parameter tuning strategy we described in Appendix E.3.
>
> **(3)** For the non-local aggregation layer number $L$, a noticeable decline in model performance is observed when $L$ exceeds 10. This is attributed to the non-local nature of our new graph, which facilitates efficient information exchange between nodes. Exceeding a certain number of layers may potentially lead to oversmoothing, where there is an overemphasis on global information, thus degrading model performance. However, **choosing the number of layers within a reasonable range generally ensures consistent and impressive model performance**, as verified in Figures 9p-9t of our revised manuscript.
>
> Sensitivity analysis of parameter $\tau$ in node classification accuracy
> |          |   1e-3  |   1e-2  |   0.1    |   0.5    |   1      |   10     |   100    |
> |----------|----------|----------|----------|----------|----------|----------|----------|
> |   Texas  |   90.98  |   91.80  |   92.95  |   94.59  |   93.61  |   91.64  |   88.52  |
> |   Cora   |   80.84  |   81.41  |   88.57  |   89.82  |   89.57  |   85.24  |   38.54  |
>
> Sensitivity analysis of parameter $\eta$
> |          |   1e-3  |   1e-2  |   0.1    |   0.5    |   1      |
> |----------|----------|----------|----------|----------|----------|
> |   Texas  |   90.00  |   91.80  |   93.11  |   94.75  |   94.59  |
> |   Cora   |   88.72  |   88.93  |   89.18  |   89.80  |   89.52  |
>
> Sensitivity analysis of parameter $\epsilon$
> |          |   0      |   1e-5  |   1e-4  |   1e-3  |   1e-2  |   0.1    |
> |----------|----------|----------|----------|----------|----------|----------|
> |   Texas  |   94.10  |   94.10  |   94.26  |   94.75  |   93.93  |   93.61  |
> |   Cora   |   89.80  |   89.72  |   89.87  |   89.69  |   89.41  |   89.11  |
>
> Sensitivity analysis of parameter $L$
> |          |   1      |   5      |   10     |   20     |   30     |
> |----------|----------|----------|----------|----------|----------|
> |   Texas  |   93.44  |   94.75  |   92.95  |   91.80  |   91.15  |
> |   Cora   |   89.06  |   89.87  |   88.77  |   86.04  |   83.61  |

---

> ### Author Response · Authors · 2023-11-22
> **As the discussion period is drawing to a close, we would be appreciative to receive your feedback.**
>
> Dear Reviewer 2DDa,
>
> We are deeply grateful for your time and effort in reviewing our paper. As the discussion period is drawing to a close, we sincerely hope that our rebuttal has carefully addressed each of your comments.
>
> In particular, we have clarified the unique position of our work in relation to prior studies on the correlation between spectral and spatial domain graph learning. Besides, we have elaborated on how our SAF framework overcomes certain limitations of current spectral GNNs. As you kindly suggested, additional experimental results, focusing on ablations and the sensitivity analysis of parameters, are also provided to enable a comprehensive analysis of SAF.
>
> If you have any other comments or questions, we are happy to engage and provide further clarification. Thank you for your invaluable feedback and attention to our submission.
>
> Best regards,
>
> 4497Authors

---

### Author Response · Authors · 2023-11-17
**Global Response From Authors**

We sincerely thank all reviewers for your thorough evaluations and insightful comments. We appreciate the positive acknowledgment our work has received in both theoretical and empirical aspects. An updated version of our manuscript, incorporating all of the reviewers' suggestions, has been submitted. In the following, we first provide a global response, followed by detailed answers to each identified weakness and question. We hope these explanations sufficiently address your concerns. Please let us know if you have any further questions.

> Concerns about Spectral Decomposition

Thanks for your insightful comments and concerns regarding the use of spectral decomposition in our SAF framework. We appreciate the opportunity to address these points and further explain the rationale behind our approach.

**(1) Practical Significance:** We acknowledge that the preprocessing time and space complexity of spectral decomposition are non-trivial aspects in the implementation of our approach. However, **it is important to note that the utility of spectral decomposition in GNNs has been increasingly recognized in recent literature.** This is evident in its application across various domains, such as graph positional encoding [1,2,3,4], graph transformer [5,6,7], spectral graph convolution [4,8], graph domain adaptation [9], and graph robustness [10]. Particularly in GNN model design, innovations like SignNet and BasisNet [4] have optimized eigenvector processing to encompass all types of spectral graph convolutions. Models like Specformer [11] and FE-GNN [12], which respectively utilize spectral and singular value decomposition, highlight the value of these approaches in enhancing model expressiveness and handling complex graph structures. In line with these advancements, our SAF framework employs spectral decomposition to create a new graph for effective non-local aggregation. We have indeed elaborated on these points in Appendix C.3 of our manuscript.

**(2) Necessity of Spectral Decomposition in SAF:** Our SAF framework pre-computes spectral decomposition once per graph and reuses it for efficient matrix inversion in Eq. (6). While it is possible to circumvent this decomposition by introducing an additional learnable polynomial function, i.e., $\hat{\mathbf{A}}_ \text{new} \approx \mathbf{U} h_ \chi(\mathbf{\Lambda}) \mathbf{U}^T = h_ \chi(\hat{\mathbf{L}})$, such an alternative would necessitate extra parameter training and repeated computation during the forward pass, possibly compromising the non-local properties we aim to capture. Despite the computational requirements, **spectral decomposition still serves an indispensable role in accurately recovering our new graph** -- a trade-off we believe is justified given the enhanced interpretability and improvements that our SAF provides.

**(3) Non-trival Design of Spectral Filter:** Access to eigenvalues and eigenvectors, while structuring the application of spectral filters to graph signals (via the explicit domain transformation $\mathbf{U}^T \mathbf{X}$), does not inherently simplify the task of designing an effective filter. **Both methods -- polynomial approximation and operations in the eigenspace -- face the significant challenge of learning adaptive spectral filters for varying graph topologies and label patterns.** This challenge persists irrespective of the knowledge of eigenvalues and eigenvectors. For polynomial-based approaches, their advantages extend beyond computational efficiency; they are particularly adept at fitting a diverse range of complex filter functions. Whereas, in our research, we strategically utilize eigenspace operations to address a specific shortfall of polynomial spectral GNNs -- their limited capability in capturing global graph information.

**(4) Addressing Computational Concerns:** In our approach, **spectral decomposition is a one-time computation per graph, enabling efficient reuse.** This aspect is crucial, as the forward-pass cost in model training often exceeds the preprocessing expense of eigendecomposition. To empirically validate this, we have compared the time and space overheads of spectral decomposition with the training of various models (see the tables provided in our subsequent response). It is evident that for most datasets, the time consumed by decomposition is significantly less than the time required for model training. For larger graphs such as Pubmed and Amazon-ratings, although the decomposition time is greater than that for local GNNs like BernNet [13] and ChebNetII [14], it is still faster than the more advanced global GNNs including SAF, NodeFormer [15], and GloGNN++ [16]. In case of space overheads, similar patterns can be observed. All these computational analysis and findings have been updated in Appendix F.5 of our revised paper.

---

> ### Author Response · Authors · 2023-11-17
> **Tables and References**
>
> >  Tables regarding time and space overheads, where local GNNs (BernNet [13] and ChebNet [14]) and global GNNs (SAF, NodeFormer[15], and GloGNN++ [16]) are included for comparison.
>
> Time overheads (s).
> |                    |   Cham.  |   Squi.  |   Texas  |   Corn.  |   Actor  |   Cora   |   Cite.  |   Pubm.    |   Amaz.    |
> |--------------------|----------|----------|----------|----------|----------|----------|----------|------------|------------|
> |   BernNet [13]     |   8.36   |   13.74  |   3.92   |   4.16   |   4.88   |   5.24   |   5.52   |   6.06     |   13.98    |
> |   ChebNetII [14]   |   22.82  |   30.73  |   11.47  |   9.64   |   14.88  |   19.96  |   16.14  |   36.91    |   22.50    |
> |   SAF              |   11.55  |   18.78  |   4.38   |   4.70   |   5.36   |   6.04   |   6.12   |   18.43    |   47.47    |
> |   NodeFormer [15]  |   58.96  |   79.66  |   14.29  |   18.89  |   66.20  |   19.25  |   32.00  |   68.57    |   122.91   |
> |   GloGNN++ [16]    |   35.63  |   68.31  |   4.47   |   3.00   |   73.13  |   32.68  |   12.35  |   5266.53  |   3614.37  |
> |   Decomposition    |   0.58   |   1.59   |   0.02   |   0.02   |   3.93   |   1.00   |   0.77   |   21.34    |   40.88    |
>
>
> Space overheads (MB).
> |                    |   Cham.  |   Squi.  |   Texas  |   Corn.  |   Actor  |   Cora  |   Cite.  |   Pubm.  |   Amaz.  |
> |--------------------|----------|----------|----------|----------|----------|---------|----------|----------|----------|
> |   BernNet [13]     |   72     |   232    |   5      |   5      |   292    |   64    |   152    |   1546   |   2389   |
> |   ChebNetII [14]   |   72     |   231    |   5      |   5      |   291    |   63    |   152    |   1584   |   2355   |
> |   SAF              |   112    |   440    |   5      |   5      |   733    |   120   |   237    |   4515   |   6966   |
> |   NodeFormer [15]  |   1522   |   3965   |   15     |   37     |   775    |   480   |   764    |   2119   |   3056   |
> |   GloGNN++ [16]    |   290    |   1525   |   5      |   5      |   2471   |   331   |   607    |   17892  |   25260  |
> |   Decomposition    |   141    |   540    |   1      |   1      |   1206   |   140   |   239    |   7641   |   11442  |
>
>
> > References
>
> [1] Belkin, Mikhail, and Partha Niyogi. "Laplacian eigenmaps for dimensionality reduction and data representation." Neural computation 15.6 (2003): 1373-1396.
>
> [2] Wang, Haorui, et al. "Equivariant and stable positional encoding for more powerful graph neural networks." ICLR, 2022
>
> [3] Dwivedi, Vijay Prakash, et al. "Benchmarking graph neural networks." arXiv preprint arXiv:2003.00982 (2020).
>
> [4] Lim, Derek, et al. "Sign and basis invariant networks for spectral graph representation learning." ICLR, 2023
>
> [5] Kreuzer, Devin, et al. "Rethinking graph transformers with spectral attention." NeurIPS, 2021.
>
> [6] Kim, Jinwoo, et al. "Pure transformers are powerful graph learners." NeurIPS, 2022.
>
> [7] Rampášek, Ladislav, et al. "Recipe for a general, powerful, scalable graph transformer." NeurIPS, 2022.
>
> [8] Liao, Renjie, et al. "Lanczosnet: Multi-scale deep graph convolutional networks." ICLR, 2019.
>
> [9] You, Yuning, et al. "Graph domain adaptation via theory-grounded spectral regularization." ICLR, 2022.
>
> [10] Chang, Heng, et al. "Not all low-pass filters are robust in graph convolutional networks." NeurIPS, 2021.
>
> [11] Bo, Deyu, et al. "Specformer: Spectral graph neural networks meet transformers." ICLR, 2023.
>
> [12] Sun, Jiaqi, et al. "Feature expansion for graph neural networks." ICML, 2023.
>
> [13] He, Mingguo, et al. "Bernnet: Learning arbitrary graph spectral filters via bernstein approximation." NeurIPS, 2021.
>
> [14] He, Mingguo, et al. "Convolutional neural networks on graphs with chebyshev approximation, revisited." NeurIPS, 2022.
>
> [15] Wu, Qitian, et al. "Nodeformer: A scalable graph structure learning transformer for node classification." NeurIPS, 2022.
>
> [16] Li, Xiang, et al. "Finding global homophily in graph neural networks when meeting heterophily." ICML, 2022.

---

### Meta-Review · Area_Chair_9qSE · 2023-12-08

**Metareview:**

In this submission, the authors proposed a spatially adaptive filtering (SAF) framework, which not only provides new insights into spectral GNNs through the lens of spatial filtering but also leads to a new method to aggregate non-local information hidden in graphs. Experiments in various node classification tasks demonstrate the potential of the proposed SAF framework.

Strengths: (1) This paper is well-written and easy to follow. (2) Comparison experiments are convincing, and more analytic experiments are added in the rebuttal phase.

Weaknesses: All reviewers and AC have concerns about the implementation of the SAF framework. Explicit eigendecomposition, obviously, limits the application of the proposed method in practice. Although the authors provided analytic experiments on the runtime and memory cost of the eigendecomposition step, this concern is not fully resolved because of the limited size of the graphs involved in the experiments. Additionally, because of using explicit eigendecomposition, AC doubts the numerical stability and robustness of the proposed method to data noise.

After discussing with the reviewers, AC suggests the authors add more analytic and experimental content to enhance the feasibility of the proposed method in practical scenarios. Resubmitting a revised version to another avenue is strongly encouraged.

**Justification For Why Not Higher Score:**

All reviewers have concerns about the feasibility of the proposed method in practical applications, and this problem is not fully resolved in the rebuttal phase.

**Justification For Why Not Lower Score:**

N/A

---

### Decision · Program_Chairs · 2024-01-16

Reject